# Long term but not short term exposure to obesity related microbiota promotes host insulin resistance

Kevin P. Foley[1], Soumaya Zlitni[2], Emmanuel Denou[1], Brittany M. Duggan[1], Rebecca W. Chan [1], Jennifer C. Stearns [3] & Jonathan D. Schertzer [1]

The intestinal microbiota and insulin sensitivity are rapidly altered after ingestion of obesogenic diets. We find that changes in the composition of the fecal microbiota precede changes in glucose tolerance when mice are fed obesogenic, low fiber, high fat diets (HFDs). Antibiotics alter glycemia during the first week of certain HFDs, but antibiotics show a more robust improvement in glycemic control in mice with protracted obesity caused by long-term feeding of multiple HFDs. Microbiota transmissible dysglycemia and glucose intolerance only occur when germ-free mice are exposed to obesity-related microbes for more than 45 days. We find that sufficient host exposure time to microbiota derived from HFD-fed mice allows microbial factors to contribute to insulin resistance, independently from increased adiposity in mice. Our results are consistent with intestinal microbiota contributing to chronic insulin resistance and dysglycemia during prolonged obesity, despite rapid diet-induced changes in the taxonomic composition of the fecal microbiota.

[1] Department of Biochemistry and Biomedical Sciences, Farncombe Family Digestive Health Research Institute McMaster University, Hamilton L8N 3Z5 ON, Canada. [2] Departments of Genetics and Medicine, Stanford University, Stanford 94305 California, USA. [3] Department of Medicine, McMaster University, Hamilton L8N 3Z5 ON, Canada. Correspondence and requests for materials should be addressed to J.D.S. (email: schertze@mcmaster.ca)

The majority of individuals with features of prediabetes, such as glucose intolerance, eventually develop T2D[1]. Environmental factors such as diet and lower activity/exercise levels contribute to the increased prevalence of pre-diabetes, which. coincides with increased incidences of obesity[2]. The composition of the intestinal microbiota can also influence postprandial glucose responses[3]. The composition of the gut microbiota is altered by obesity, T2D, insulin sensitizing drugs, age, diet constituents and exercise among other environmental factors[4–8]. It is not clear how or when the microbiota contributes to the progression of glucose intolerance versus obesity. This is an important distinction because not all obese individuals develop insulin resistance, glucose intolerance or T2D. Further, the timing and progression of obesity can be different from that of dysglycemia

The intestinal microbiota can contribute to host energy balance and lipid deposition through hormonal cues[9]. Microbial transplant experiments show that differences in the gut bacterial community are sufficient to increase adiposity, independently of host genetics[10,11]. Hence, there is a strong connection between the microbiota and obesity, but it is not clear if a microbiota-induced change in adiposity is the primary factor contributing to glucose intolerance. In mice, diet influences the composition of the gut microbiota more than host genetics and it is already known that increasing dietary fat content can rapidly perturb the composition of the microbiota within days[12]. An obesogenic, low fiber, high-fat diet (HFD) is a widely used model that induces insulin resistance and glucose intolerance that is coincident with obesity in rodents. The mechanisms underpinning glucose intolerance during the initial stages of feeding this obesogenic diet are different than those governing chronic glucose intolerance during prolonged diet-induced obesity. For example, ectopic lipid accumulation in the skeletal muscle and liver is associated with glucose intolerance after the first few days of HFD feeding, whereas metabolic tissue inflammation plays a more prominent role in propagating glucose intolerance after months of HFD-feeding in mice[13]. It is not clear if diet-induced changes in the microbiota contribute to the mechanisms underpinning acute versus chronic insulin resistance and glucose intolerance. We wanted to fill this knowledge gap by testing if short term or long term diet-induced changes in the microbiota were sufficient to alter glycemic control.

We find that changes in the composition of the microbiota precede overt dysglycemia in mice fed two separate obesogenic diets. Only long-term feeding of an obesogenic diet promotes transmissible glucose intolerance, which can occur independently of changes in adiposity. Long-term exposure of the host to the rapidly changed microbiota is necessary and sufficient for bacteria to contribute to poor glucose control. Our results support a model where sufficient exposure time of the host to the microbiota-derived factors present during an obesogenic diet is a factor that permits microbes to contribute to poor glucose control. Our data support the concept that host exposure time is a key factor to consider in the development of dysglycemia and warrant caution in the assumption that continual evolution of the microbiota during long-term feeding of an obesogenic diet is required for poor host glucose control. This time required for microbe factors to promote dysglycemia should be considered independent from obesity and despite rapid diet-induced changes in the microbiota.

## Results

**Obesogenic diets cause glucose intolerance within 4 days**. We used obesogenic diets (Research Diets, D12451 and D12492) that contain either 45 or 60% kcal derived from fat (i.e., lard and soybean oil). Both of these obesogenic diets have ~6% fiber. The chow diet contained 13% fiber and 17% fat content (Harlan Teklad 8640). Glucose tolerance was not different after 1 day of feeding two separate obesogenic diets compared to the chow diet (Fig. 1a). Four days of feeding either obesogenic diet marked the first time that both 45% and 60% HFD caused glucose intolerance compared to mice fed a chow diet (Fig. 1b). Glucose intolerance persisted after 14 days (Fig. 1c) and 14 weeks of feeding 45 or 60% HFD (Fig. 1d). The 60% HFD caused worse glucose control compared to the 45% HFD, when tested between 4 days and 14 weeks of feeding (Fig. 1b–d).

**Increased adiposity precedes dysglycemia during HFD feeding**. One day of feeding either obesogenic diet increased body mass compared to chow-fed mice (Fig. 1e). Between 4 and 7 days of HFD-feeding, a 60% HFD caused a greater change in body mass compared to a 45% HFD (Fig. 1e). One day of feeding 60% HFD increased body fat percentage, whereas it took 3 days of feeding a 45% HFD to increase adiposity (Fig. 1f). Between 3 and 7 days of HFD feeding, a 60% HFD increased adiposity more than a 45% HFD (Fig. 1f). One day of feeding either obesogenic diet transiently increased food consumption, but food consumption (per gram of food) was lower for both obesogenic diets from 4 days up to 2 weeks (Fig. 1g). These results show that: (1) a single day of feeding an obesogenic diet increases body mass and body fat percentage and (2) the obesogenic diet that contains a higher energy content per gram of food (i.e., 60% HFD) increases adiposity to a greater extent during the first week of feeding obesogenic diets.

**Microbiota changes precede dysglycemia during HFD feeding**. Principal coordinate analysis (PCoA) of fecal bacterial 16s rRNA gene profiles showed that the bacterial community did not appreciably change during 0–7 days of feeding mice a chow diet (Fig. 2a, top). When compared to the pre-diet gut microbial communities, 1–7 days of feeding either a 45 or 60% HFD altered the fecal bacterial community (Fig. 2a, middle and bottom). A direct comparison of diets on day 3 of feeding (i.e., before detectable changes in glucose tolerance) showed that mice fed either obesogenic diet clustered together and separately from chow fed mice (Fig. 2b). A snapshot of the 12 most abundant taxonomic assignments at the genus level across the first week of each diet illustrates the rapid change in the relative composition of the fecal microbiota that both obesogenic diets cause within 1 day and the similarity of the most abundant taxa between day 1 and day 7 of changing from a chow diet to an obesogenic diet (Fig. 2c). These results are consistent with previous reports documenting the rapid effect of diet on gut microbial taxa[12].

We then focussed on the 3rd day of changing diets, since this time-point of feeding either obesogenic diet preceded overt glucose intolerance and because PCoA analysis showed an altered microbial community. A total of 227 genus level taxa were detected. Compared to a chow diet, we found that 36 taxa were different after 3 days of feeding either a 45 or 60% HFD when the relative abundance of all diets were compared by non-parametric statistical analysis with multiple hypothesis correction ($p$-value < 0.05). These data were then plotted on a heat map of fold changes compared with chow diet for only the statistically significant taxa altered by 3 days of feeding either obesogenic diet. These data show that the same taxa were altered by both obesogenic diets (45% HFD and 60% HFD) by 3 days of feeding (Fig. 2d). The stability of these changes can be observed in Supplementary Figure 1, which shows fold changes in these 36 taxa over the first 7 days of all diets normalized to day 0 (i.e., a chow diet). Most taxa showed increases or decreases, within 1 day of starting an obesogenic diet, which persisted over the first 7 days

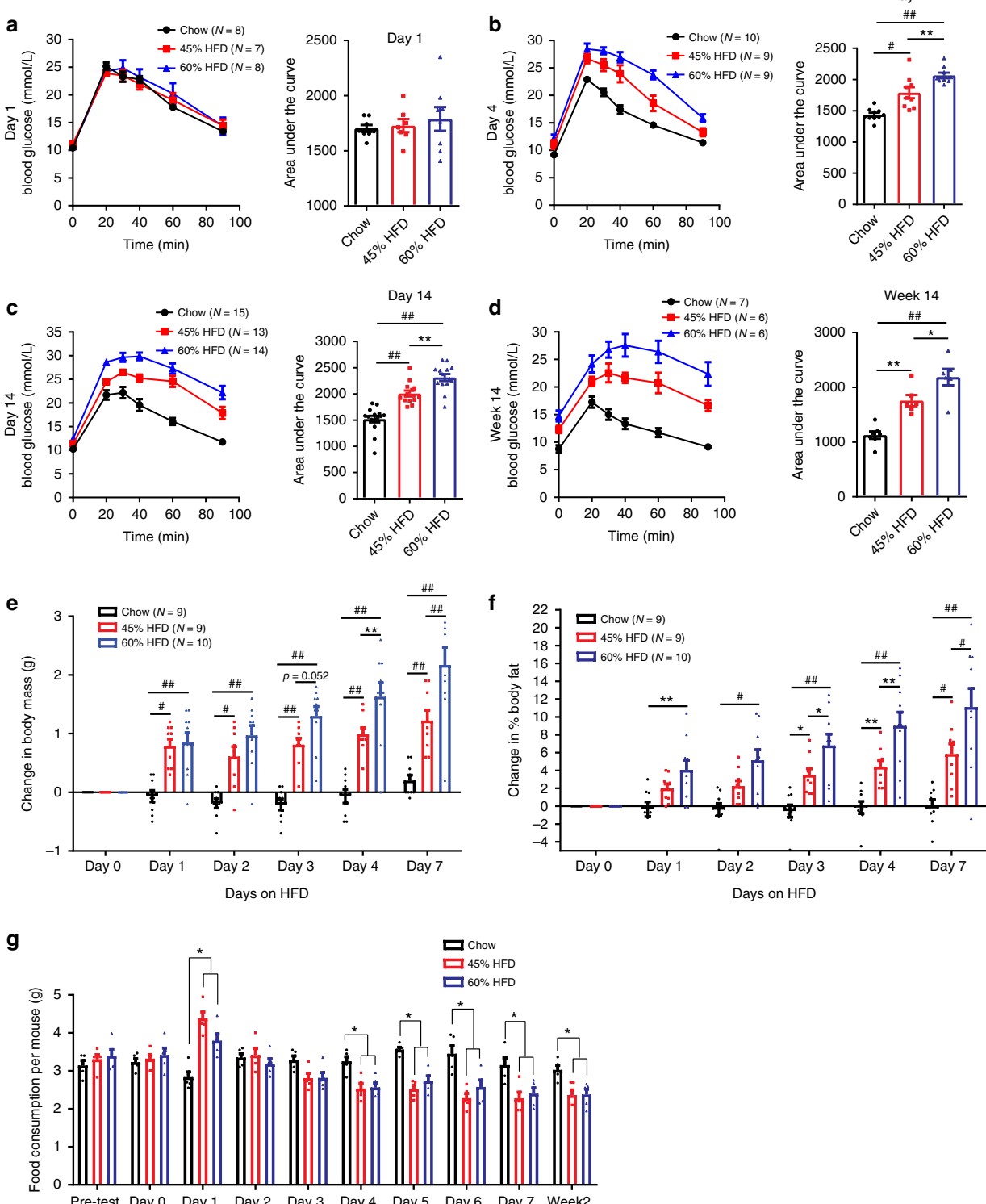

**Fig. 1** Four days of an obesogenic diet is sufficient to induce glucose intolerance and increase adiposity in mice. **a–d** Mice were fed a chow, 45% HFD, or 60% HFD for 1 day (**a**: N = 8, 7, 8), 4 days (**b**: N = 10, 9, 9), 14 days (**c**: N = 15, 13, 14), or 14 weeks (**d**: N = 7, 6, 6) then tested for glucose tolerance with a 2 g per kg (**a–c**) or 0.9 g per kg (**d**) glucose dose by *i.p.* injection. Blood glucose measures were taken at indicated time points, which are shown in the glucose tolerance test (GTT) curve and used to calculate the area under the curve (AUC). Statistical significance was measured as *p* < 0.05 using one-way ANOVA. Post hoc analysis was performed using Tukey's multiple comparisons test (**p* < 0.05; ***p* < 0.01; #*p* < 0.001; ##*p* < .0001). Change in body weight (**e**) and change in % body fat (**f**) were calculated as the difference between Day 0 and subsequent days within each animal. Body weight and adiposity (% body fat) were measured in mice fed a chow, 45% HFD, or 60% HFD for 7 days (N = 9, 9, 10). Statistical significance was measured as *p* < 0.05 using two-way ANOVA with repeated measures (time). Post Hoc analysis was performed using Tukey's multiple comparisons test (**p* < 0.05; ***p* < 0.01; #*p* < 0.001; ##*p* < .0001). **g** Food consumption (N = 5 cages per group) was measured daily and expressed per mouse. Statistical significance was measured as *p* < 0.05 using one-way ANOVA for each time point. Post hoc analysis was performed using Tukey's multiple comparisons test (**p* < 0.05). All values are mean ± SEM

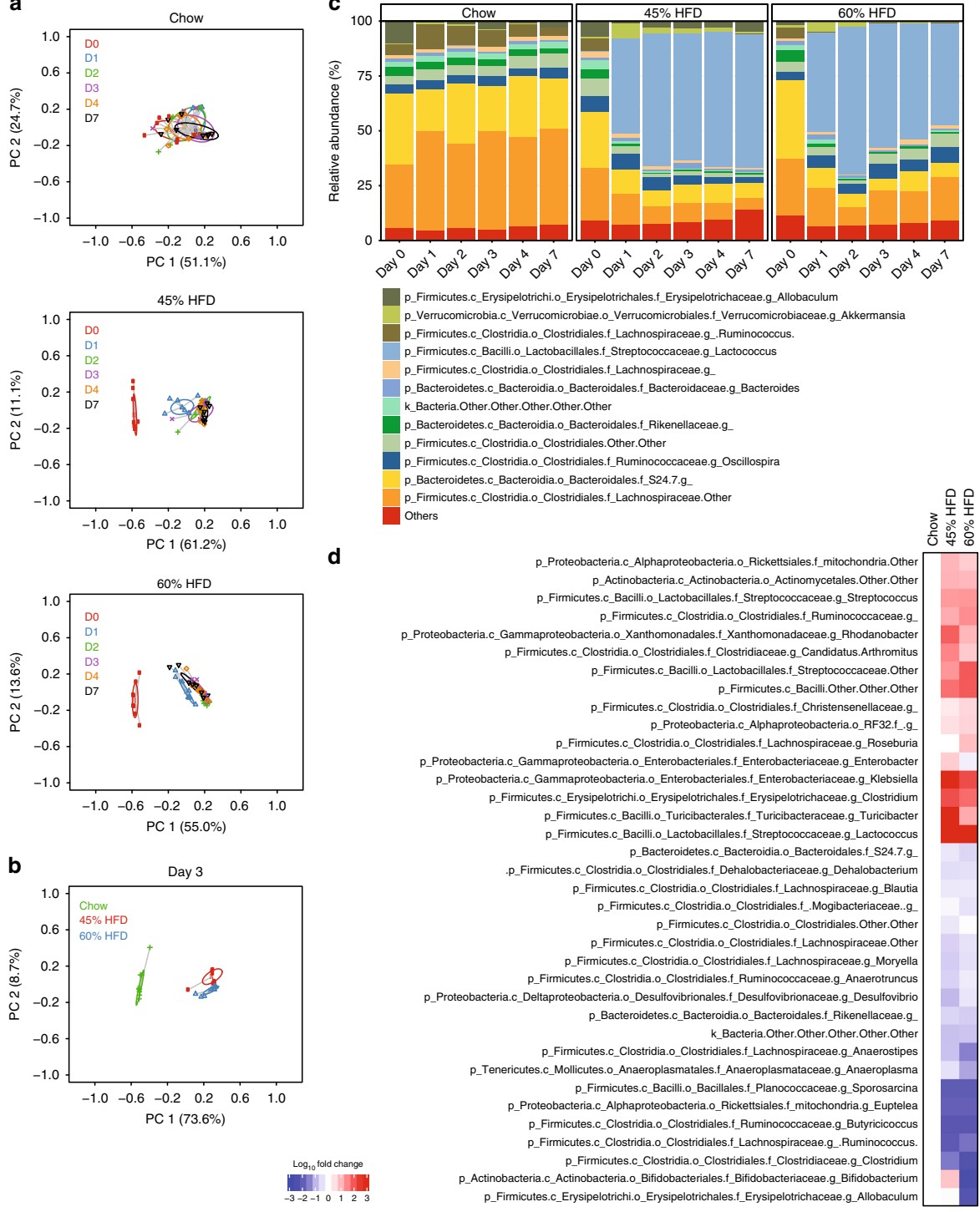

**Fig. 2** Obesogenic diet feeding changes the fecal microbiota, which precedes glucose intolerance in mice. Mouse fecal samples were taken over the first 7 days (D0–D7) of feeding obesogenic, low fiber, high fat diets (HFD) and processed for bacterial DNA sequencing ($N = 7$–8). All mice were on a Chow diet on Day 0 (D0). **a** PCoA of Bray-Curtis dissimilarity for all samples over the 7 days on Chow diet, 45% HFD, or 60% HFD. **b** PCoA of Bray-Curtis dissimilarity after 3 days of eating each of the 3 diets. **c** Stacked bar graph showing the relative abundance of the 12 most abundant bacterial taxa (Genus level) over the first 7 days of eating each of the 3 diets. **d** Heat map of the 36 microbial taxa that were significantly different 3 days after eating Chow, 45% HFD, and 60% HFD. Non-parametric analysis of variance for each taxon between groups was conducted using the Kruskal–Wallis test. Taxa that passed the significance threshold of $p < 0.05$ were analyzed using the pairwise Wilcoxon rank sum test. Correction for multiple hypothesis testing (FDR) was calculated using the Benjamini-Hochberg method. Statistical significance was accepted at $p < 0.05$. Fold change in relative abundance of the taxa that significantly changed between the groups was expressed relative to Chow and plotted in the heatmap

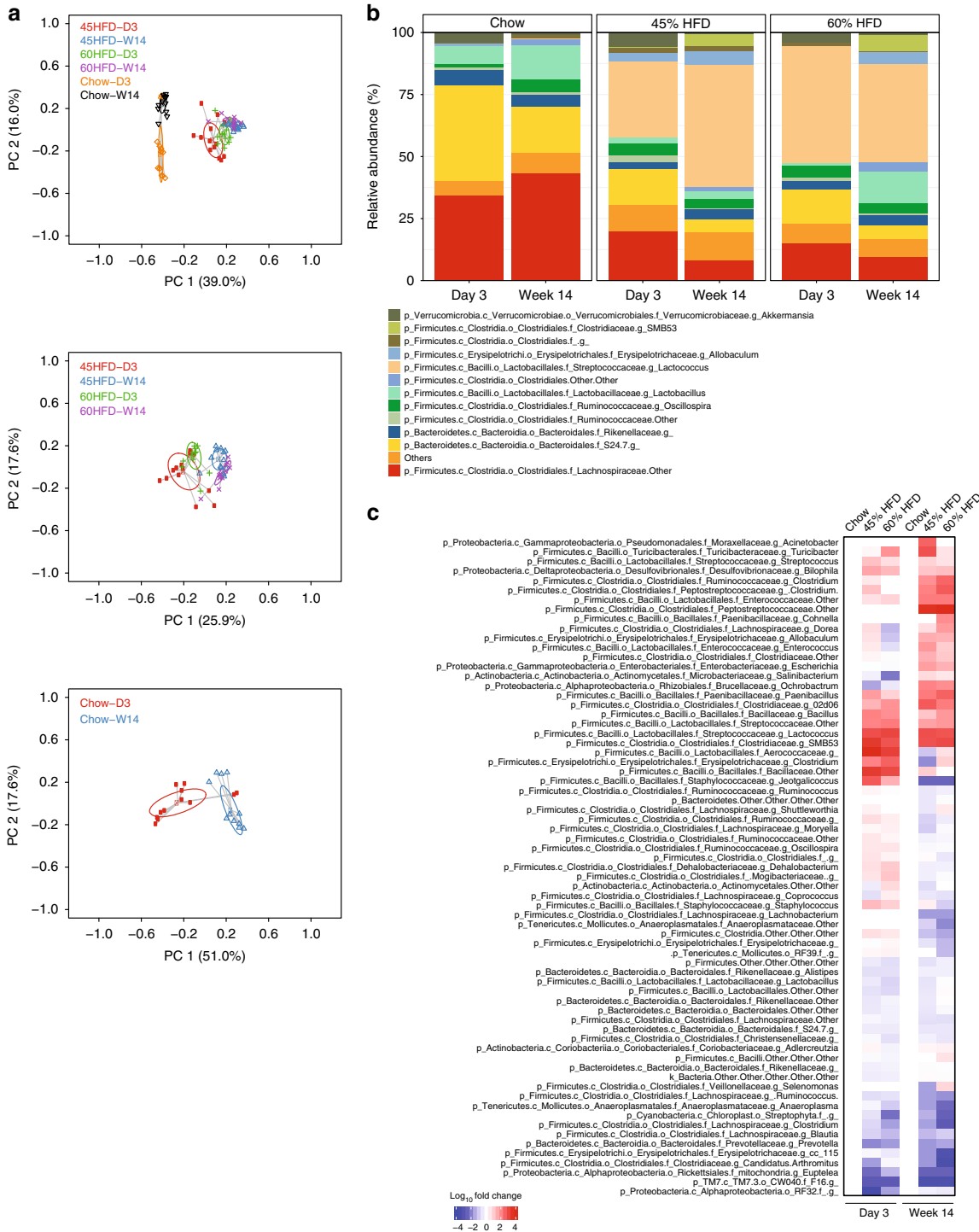

**Fig. 3** Rapid changes in the composition of the fecal microbiota are maintained during prolonged obesogenic diet feeding. Fecal samples were taken 3 days (D3) and at 14 weeks (W14) after feeding chow or each obesogenic diet and processed for bacterial DNA sequencing (Chow = 12, 45% HFD = 13, 60% HFD = 12 mice). **a** PCoA of Bray-Curtis dissimilarity between each of the 3 diet groups (Day 3 and Week 14) (top panel), between the two obesogenic diet groups only (Day 3 and Week 14) (middle panel), and between Day 3 and Week 14 of mice that only ate the chow diet (bottom panel). **b** Stacked bar graph showing the relative abundance of the 12 most abundant bacterial taxa (Genus level) at Day 3 and Week 14 mice fed each diet. **c** Heat map of the 69 microbial taxa that differed between mice fed Chow, 45% HFD, and 60% HFD on Day 3 or Week 14. The average relative abundances of each taxon detected in mice fed the respective diets for 3 days or 14 weeks were compared between groups. For each time point (Day 3 or Week 14), non-parametric analysis of variance for each taxon between the diet groups was conducted using the Kruskal–Wallis test. Taxa that passed the significance threshold of $p < 0.05$ were analyzed using the pairwise Wilcoxon rank sum test. Correction for multiple hypothesis testing (FDR) was calculated using the Benjamini-Hochberg method. Statistical significance was accepted at $p < 0.05$. Fold change in relative abundance of the taxa that significantly changed between the groups was expressed relative to Chow within each timepoint and plotted in the heatmap

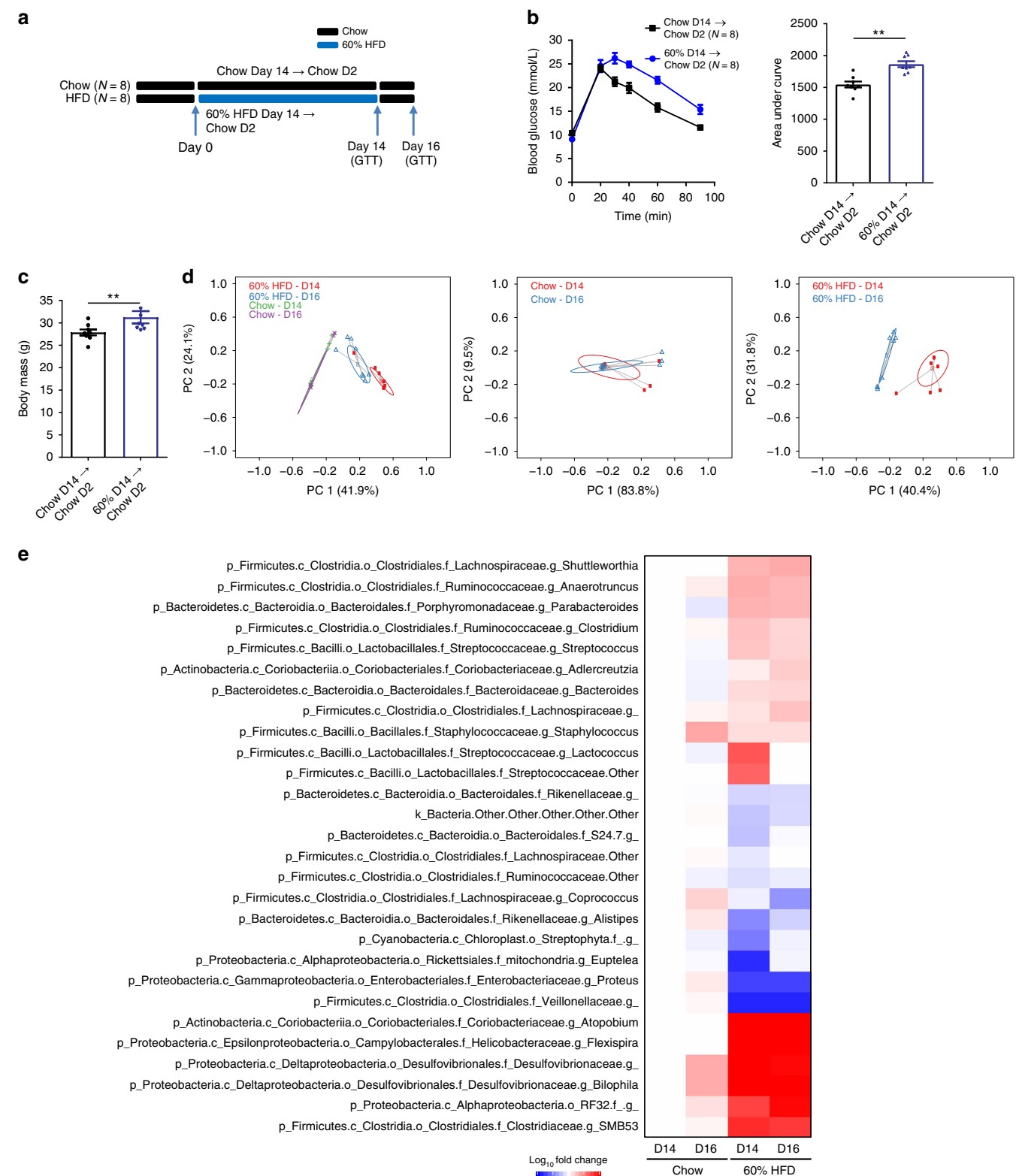

of feeding (Supplementary Figure 1). Furthermore, the low day to day variability in taxa abundance can be seen in the chow fed mice.

We next directly compared short-term changes in the microbiota (day 3 of feeding obesogenic diets) to long-term changes in the microbiota (week 14 of feeding obesogenic diets). We found that an obesogenic diet, irrespective of the length of diet, was the major factor in altering the composition of the fecal microbial community, where mice fed a chow diet clustered together and separately from either HFD diet fed for 3 days or 14 weeks (Fig. 3a, top panel). We calculated the effect of each diet, time and the interaction of diet and time on the variance seen in the microbiota with a PERMANOVA. We found an effect of diet and an effect of time on the microbiota ($p$-value = 0.001) as well

**Fig. 4** Glucose intolerance persists despite rapid changes in fecal microbiota after removing an obesogenic diet for 2 days. Mice were fed chow or 60% HFD ($N = 8$, 8) for 14 days before HFD was replaced with a chow diet for 2 days (Day 16) (**a**). **b** Glucose tolerance test (GTT) curve with area under the curve (AUC) (2 g per kg glucose, *i.p.*) and **c** body mass on day 16. Statistical significance was measured as $p < 0.05$ using Student $t$-test (*$p < 0.05$; **$p < 0.01$; #$p < 0.001$; ## $p < .0001$). Values are mean ± SEM. **d** PCoA of Bray-Curtis dissimilarity for Chow (Day 14), Chow (Day 16), 60% HFD (Day 14), and 2 days HFD removal (60HFD Day 16). PCoA plots for all groups of mice (left panel), mice only fed a chow diet (middle panel), and mice that were fed a 60% HFD with or without replacement of the obesogenic diet with a chow (right panel). **e** Genus level changes in the microbiota relative to 14 days of chow diet. The average relative abundance of each taxon detected in fecal samples was compared across the different groups of mice. Non-parametric analysis of variance for each taxon between the treatment groups (Chow day 14, Chow Day 16, 60% HFD Day 14, 60% HFD Day 16) was conducted using the Kruskal–Wallis test. Taxa that passed the significance threshold of $p < 0.05$ were analyzed using the pairwise Wilcoxon rank sum test. Correction for multiple hypothesis testing (FDR) was calculated using the Benjamini-Hochberg method. Statistical significance was accepted at $p < 0.05$. Fold change in relative abundance of the taxa that significantly changed within either diet group was expressed relative to Chow Day 14 and plotted in the heatmap

as an interaction between diet and time ($p$-value $= 0.001$). Diet explained 5.5% of the variation in the microbiota, whereas time explained 3.2% and the interaction of diet and time explained an additional 4.5% of the variation in the microbiota. The interaction effect did not decrease the effect of diet or time alone, which suggests that both HFDs have a different effect over time on the microbiota compared with the effect over time in mice on a chow diet (Fig. 3a, middle panel). The change we observed in the microbiota over time in chow fed mice illustrates an age-related change that is independent of diet. A snapshot of the most abundant taxa at the genus level showed that either obesogenic diet (45 or 60% HFD) altered the average relative microbial composition in the feces compared to mice fed a chow diet and also showed the (14 week) age-related shift over time that we saw in control animals (Fig. 3b). In particular, *Akkermansia* is reduced from Day 3 to Week 14 in all groups.

We then focussed on directly comparing the fecal microbiota after 3 days and 14 weeks of changing diets. A total of 205 genus level taxa were detected. We found that the abundance of 69 taxa were different after 3 days or 14 weeks of feeding either a 45 or 65% HFD, when compared to a chow diet ($p < 0.05$). Most of the specific taxa that were increased or decreased by a 45% HFD had a similar direction of change after feeding a 60% HFD (Fig. 3c). Changes in taxa abundance seen at 3 days of feeding an obesogenic diet became more prominent after 14 weeks of feeding an obesogenic diet. A small number of exceptions include members of the Firmicutes phylum that were discordant between 3 days and 14 weeks of feeding an obesogenic diet (Fig. 3c). The relative abundance of statistically different taxa on day 3 and week 14 for all diets is depicted in Supplementary Figure 2.

We next analyzed glucose tolerance and microbiota composition after short-term removal of the HFD and replacement with a chow diet by feeding mice a 60% HFD for 14 days, then switching the mice to a chow diet for 2 days (i.e., Day 16) (Fig. 4a). Removing the 60% HFD for 2 days did not significantly reduce body mass. HFD-fed mice weighed 33.4 g compared to 31.2 g after HFD feeding plus 2 days of chow diet. Despite HFD removal, the mice previously fed 60% HFD remained glucose intolerant and still had higher body mass when compared to age-matched chow-fed mice (Fig. 4b, c). PCoA analysis showed that mice fed a chow diet over the study period clustered together and distinctly from that of mice fed a 60% HFD for 14 days (Fig. 4d). It was evident that removal of the HFD for 2 days caused a distinct clustering of mice in the PCoA analysis that was intermediate between the chow fed and HFD fed mice (Fig. 4d). To further probe this we first confirmed that mice fed a chow diet for 14 and 16 days did not have distinct clustering by PCoA analysis (Fig. 4d, middle panel). However, it was clear that removing the HFD for 2 days caused a distinct PCoA profile compared to the same mice fed a HFD for 14 days (Fig. 4d, right panel). We found that 28 taxa (out of 157 detected) were significantly different (FD corrected, $p < 0.05$) when 14 days of

feeding a 60% HFD or removing this HFD diet for 2 days (i.e., D16) was compared to a chow diet for 14 days. A heat map that depicts fold change in the abundance of each taxon that significantly changed relative to chow diet feeding (day 14) shows that a small number of taxa might explain differences in the PCoA analysis after 2 days of removal of an obesogenic diet (Fig. 4e). Compared to 14 days of 60% HFD feeding, the majority of taxa were not altered after returning to chow feeding for 2 days (Fig. 4e). The genus *Lactococcus* and another taxa in the *Streptococcaceae* family were more abundant on day 14 of HFD-feeding, an effect that was rapidly lost after returning mice to a chow diet (Fig. 4e). Further, taxa in the order *Streptophyta* and genus *Euptelea* were less abundant on day 14 of HFD-feeding, an effect that was rapidly lost after returning mice to a chow diet for 2 days (Fig. 4e). The relevance of these results should be analyzed with caution and it should be carefully considered if some of these results may reflect microbial DNA in the ingested diet. A direct comparison of the relative abundance of differentially abundant taxa on day 14 of feeding a 60% HFD and after 2 days of returning these mice to a chow diet (i.e., day 16) are depicted in Supplementary Figure 3. Overall, these data show that diet-induced changes in the fecal microbiota precede changes in glucose tolerance at both the onset and removal of an obesogenic diet.

**Antibiotics improve glycemia during prolonged HFD feeding.**
We next used antibiotics to mitigate diet-induced changes in the microbiota in order to test if short-term changes in microbes correspond with altered glycemia. We have previously established an antibiotic cocktail that causes profound changes in the gut microbiota and attenuates insulin resistance after prolonged HFD-feeding[14], which has also been described by others[15]. The conditions for this experiment were based on our results showing that 4 days of HFD-feeding was sufficient to cause glucose intolerance (Fig. 1b). Mice were treated with antibiotics (0.5 g/L neomycin and 1.0 g/L ampicillin in the drinking water), which commenced 3 days prior to feeding a 60% HFD and continued for 4 days of HFD feeding (Fig. 5a). Despite small reduction in body mass and fat mass gains, antibiotics did not prevent increased adiposity during this short-term 60% HFD-feeding (Fig. 5b). This 7 day antibiotic treatment did not prevent the increase in fasting blood glucose (Fig. 5c) or glucose intolerance (Fig. 5d) induced by feeding a 60% HFD for 4 days.

We next tested if antibiotics had a similar effect during short-term feeding of a chow diet or 45% HFD (Fig. 5e). Antibiotics did not alter body mass or change in % body fat of chow-fed mice (Fig. 5f). However, antibiotics did significantly lower body mass in 45% HFD mice without altering the change in % body fat during 4 days of feeding this obesogenic diet (Fig. 5f). Antibiotics lowered fasting blood glucose in both chow-fed and 45% HFD fed mice (Fig. 5g). Similarly, antibiotics improved glucose tolerance during a glucose tolerance test in both chow-fed and 45% HFD

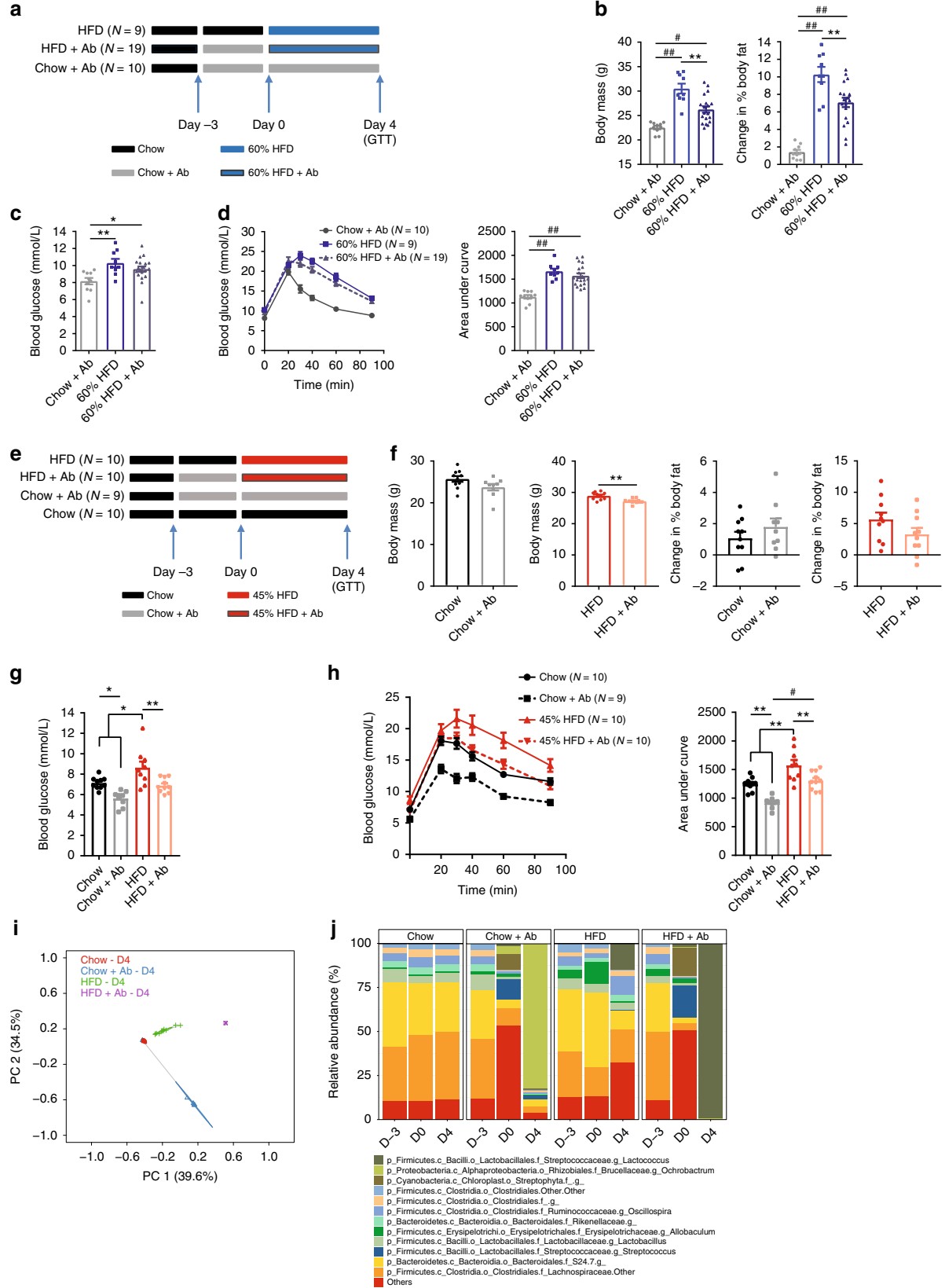

fed mice (Fig. 5h). As expected, antibiotics had a profound effect on the composition of the microbiota in mice fed a chow diet or a 45% HFD. PCoA analysis and a snapshot of the average relative abundance of the most prevalent genus level taxa showed that a change to an obesogenic diet and antibiotic exposure altered the composition of the microbiota (Fig. 5i–j). These data show that antibiotics can alter glycemic control during short term feeding of certain obesogenic diets, which may be linked with changes in body mass of mice. However, the effect of antibiotics on the worsening of glycemic control due to the short term feeding of

**Fig. 5** Antibiotics can improve glucose tolerance in mice fed chow or certain obesogenic diets for short durations. Mice were treated with or without antibiotics (1 mg/mL ampicillin and 0.5 mg/mL neomycin) in the water for 3 days before being placed on chow, 45% HFD, or 60% HFD with or without antibiotics for an additional 4 days. **a** Experimental design for testing antibiotics during the obesogenic diet containing 60% fat (Chow + Ab = 10; HFD = 9; HFD + Ab = 19). **b** Body mass and change in % body fat on Day 4 of HFD feeding with or without antibiotics. **c** Fasting blood glucose and **d** Glucose tolerance test (GTT) (2 g per kg glucose *i.p.*) and area under the curve (AUC) with and without antibiotics. **e** Experimental design for testing antibiotics during the obesogenic diet containing 45% fat (Chow = 10; Chow + Ab = 9; HFD = 10; HFD + Ab = 10). **f** Body mass and % change in body fat measures for chow fed and 45% HFD mice on Day 4 of feeding. **g** Fasting blood glucose and **h** GTT (2 g per kg glucose *i.p.*) and AUC with and without antibiotics. **i** PCoA of Bray-Curtis dissimilarity for all groups on Day 4 of chow diet or 45% HFD, with or without antibiotics. **j** Stacked bar graph showing the relative abundance of the 12 most abundant bacterial taxa (Genus level) over the entire course of the experiment in chow and 45% HFD-fed mice. Statistical significance was measured as $p < 0.05$ using Student $t$-test or one-way ANOVA. Post hoc analysis was performed using Tukey's multiple comparisons test (*$p < 0.05$; **$p < 0.01$; #$p < 0.001$; ##$p < .0001$). Values are mean ± SEM

obesogenic diets with a high energy content (i.e., 60% HFD) were not statistically different in mice.

We next used this same antibiotic regimen to change the microbiota of mice fed obesogenic diets for a longer period of time. First, mice were fed either a 60% HFD or 45% HFD for 13 weeks followed by 7 days with or without antibiotics (Fig. 6a). Mice that received 1 week of antibiotics showed improved glucose tolerance (Fig. 6b) and lower fasting blood glucose without changes in body mass (Fig. 6c) after prolonged feeding of 60% HFD. Similar effects of antibiotics improving glycemic control also occurred in mice fed a 45% HFD for 13 weeks, where mice exposed to antibiotics for 7 days had lower fasting blood glucose and improved glucose tolerance despite no change in body mass or body fat percentage (Fig. 6d–f). PCoA analysis and a snapshot of the average relative abundance of the most prevalent 12 taxa (genus level) of mice fed a 45% HFD also showed that antibiotic exposure caused profound changes in the composition of the microbiota (Fig. 6g–h). Overall, these results show that changing the microbiota with a specific antibiotic cocktail attenuates glucose intolerance and hyperglycemia after long term feeding of both obesogenic diets, but not necessarily after short-term HFD-feeding. The ability of antibiotics to improve glucose control in mice during short term feeding of 45% HFD mice was associated with a small, but significant decrease in body mass during 1 week of antibiotic treatment (day 4 of HFD). Antibiotics consistently improved glucose control during long-term feeding for both obesogenic diets, which occurred independent from changes in body mass or obesity. We now have the opportunity to mine potential differences in the constituents of the microbiota in mice where antibiotics had discordant effects on glycemia. However, we interpret the key result of these experiments to be that the contribution of the microbiota to dysglycemia either requires long term exposure to the microbes associated with obesity or long term exposure to the obesogenic diet. Rather than generate associations by mining the taxonomy of antibiotic treated mice, we next directly tested the diet-microbe exposure duration relationship to glycemia by exposing germ-free mice to specific microbe communities.

**Prolonged exposure to HFD microbiota worsen glucose control**. We first colonized germ-free mice to test if short-term HFD-induced changes in the microbiota are sufficient to promote dysglycemia. We initially tested the cumulative effect of short-term diet-induced changes in microbiota over the first 6 weeks by continually exposing chow-fed germ-free mice to the feces from donor mice fed a 60% HFD or chow diet (Fig. 7a). Germ-free mice that received daily feces from 60% HFD mice or chow fed mice had similar glucose tolerance, fasting blood glucose, and percent body fat on day 4 (Fig. 7b) after microbiota exposure. However, after 45 days of microbiota exposure germ-free mice that received feces from 60% HFD mice were more glucose

intolerant and had higher adiposity (i.e., % body fat) compared to germ-free mice that received feces from chow fed mice (Fig. 7c). PCoA analysis of all donor and recipient mice showed a distinct clustering of donor mice that ate a 60% HFD compared to all mice that ate a chow diet (Fig. 7d, left panel). This indicates that ingested diet is the major driver of changes in the microbial community of the fecal microbiota as opposed to transfer of differences in the microbiota from donors to recipients. Nevertheless, PCoA analysis comparing only the chow fed, germ-free recipient mice after 45 days of exposure to donor feces showed a distinct clustering of the mice that received HFD versus chow feces (Fig. 7d, right panel). A direct comparison of relative abundance showed that there were 9 taxa at the genus level that were significantly different after 45 days of 60% HFD feces exposure compared with exposure to chow-fed donor mouse feces (Fig. 7e). The criterion for inclusion in this analysis was that the bacterial taxa in question must be present in two-thirds of the recipient mice that were exposed to feces from chow fed or 60% HFD-fed mice. It is noteworthy that the genus *Akkermansia* was less abundant in the mice that were exposed to fecal material from HFD-fed mice (Fig. 7e). To probe the success of the fecal microbiota transfer we used an Upset plot to evaluate how many taxa were shared between donor (D) and recipient (R) mice (Fig. 7f). There were approximately 60 unique taxa within these mice (bottom left bars) and we found that 38 taxa were shared between all D and R mice (main bar graph, far right). Eleven taxa were found only in a single group whereas 6 taxa were found in all groups of mice except the mice that were eating an obesogenic diet (i.e., the HFD_D group). There were 4 taxa that were found in both donor groups but in only one of the recipient groups. These data suggest that the majority of the microbial population is shared amongst all mice and that a majority of taxa were successfully transferred to recipient groups. It is likely that the differences in relative abundance of a few key taxa are driving phenotypic changes.

Finally, we colonized germ-free mice with the feces from donor mice that had been fed a HFD for over 2 months and tested the effects on glycemia (Fig. 8a). Mice that were used as microbiota donors were fed a chow or 60% HFD for 4 weeks prior to the experiment in order to discern if microbiota transmissible dysglycemia is due to: (1) long term exposure of donor mice to obesogenic diet or (2) long term exposure of recipient mice to the microbes associated with diet-induced obesity. Germ-free, recipient mice that were all fed a chow diet, but received daily feces from chow-fed or HFD-fed donors, had similar body mass, percentage body fat, and glucose tolerance when exposed to the microbial communities for 4 days (Fig. 8b). However, chow-fed, germ-free recipient mice had increased glucose intolerance after exposure to HFD-fed donor feces for 45 days (Fig. 8c). This impaired glucose tolerance occurred despite no change in adiposity (Fig. 8c). PCoA analysis of all donor and recipient mice showed a distinct clustering of donor mice fed a 60% HFD

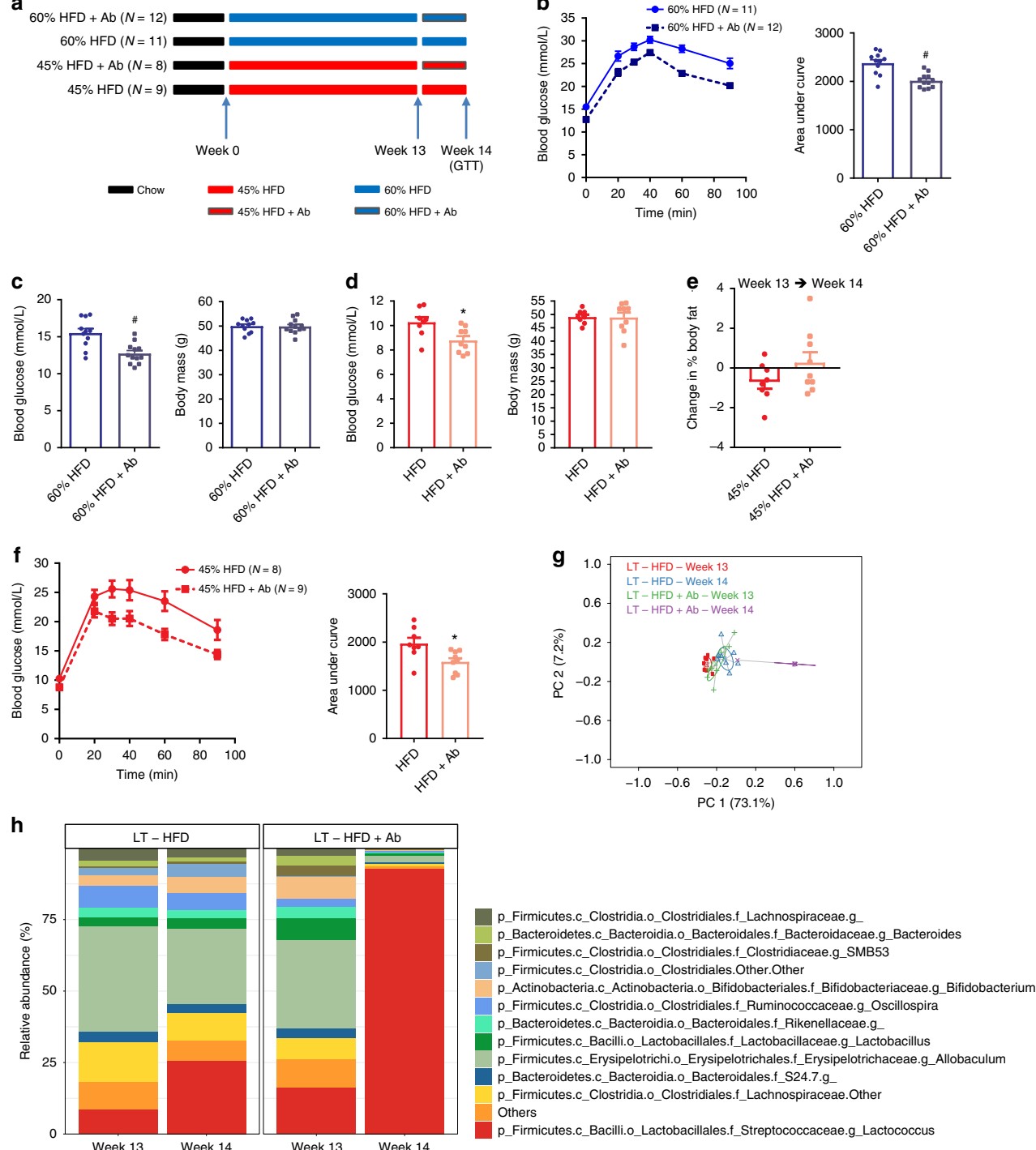

**Fig. 6** Antibiotics consistently improve glucose tolerance in mice fed obesogenic diets for long durations. **a** After 13 weeks of feeding obesogenic HFD diets, mice were treated with or without antibiotics (1 mg/mL ampicillin and 0.5 mg/mL neomycin) in the drinking water for 1 week (60% HFD = 11; 60% HFD + Ab = 12; 45% HFD = 8; 45% HFD + Ab = 9). **b** Glucose tolerance test (GTT) (2 g per kg glucose *i.p.*) and area under the curve (AUC) in mice fed a 60% HFD with and without antibiotics. **c**, **d** Fasting blood glucose and body mass in mice fed 60% HFD (**c**) and 45% HFD (**d**) with and without antibiotics. **e** Changes in body fat % in mice fed 45% HFD with and without antibiotics. **f** GTT (2 g per kg glucose *i.p.*) and AUC in mice fed a 45% HFD with and without antibiotics. **g** PCoA of Bray-Curtis dissimilarity for all mice fed a 45% HFD with and without antibiotics. **h** Stacked bar graph showing the relative abundance of the 12 most abundant bacterial taxa (Genus level) for all mice fed a 45% HFD with and without antibiotics. Statistical significance was measured as $p < 0.05$ using Student $t$-test. Post hoc analysis was performed using Tukey's multiple comparisons test (*$p < 0.05$; **$p < 0.01$; #$p < 0.001$; ##$p < .0001$). Values are mean ± SEM

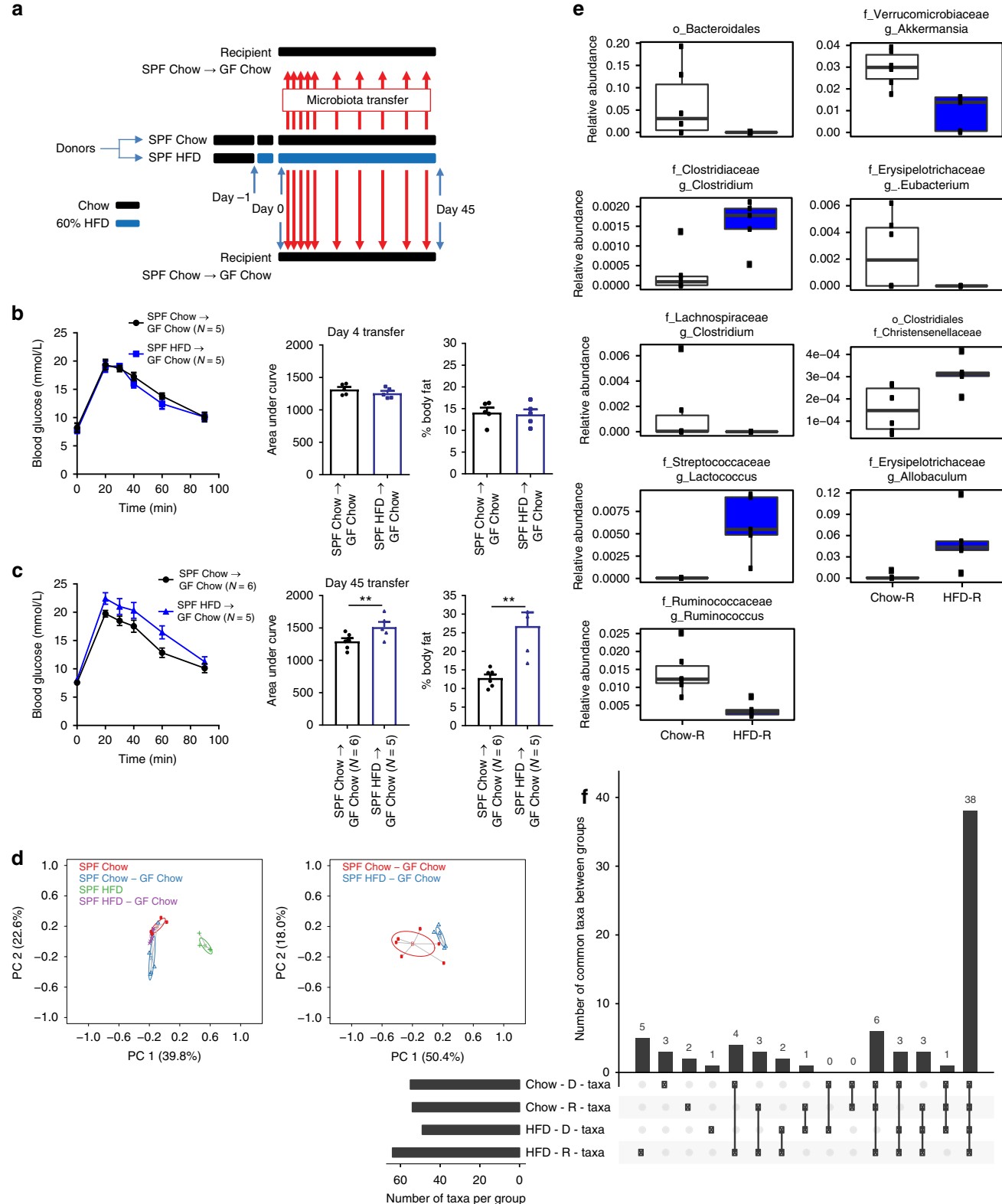

compared to all mice that ate a chow diet (Fig. 8d, left panel). Similar to previous experiments, this result indicates that ingested diet is the major driver of changes in the microbial community of the fecal microbiota. PCoA analysis comparing only the chow fed, germ-free recipient mice after 45 days of exposure to the two different types of donor feces showed a distinct clustering of the mice that received HFD versus chow feces (Fig. 8d, right panel). A

direct comparison of relative abundance revealed 5 statistically different taxa after 45 days of exposing germ-free mice to feces from 60% HFD versus chow-fed donor mice (Fig. 7e). The criterion for inclusion in this analysis was that the bacterial taxa in question must be present in two-thirds of the recipient mice that were exposed to feces from chow fed or 60% HFD-fed mice. We again used an Upset plot to test how many taxa were shared

**Fig. 7** Gut microbiota from long-term, but not short-term HFD-fed mice causes glucose intolerance in germ-free mice. **a** Schematic of experimental design. Specific pathogen free (SPF) donor mice were placed on chow or 60% HFD on Day -1. On Day 0, and each subsequent day, feces were transferred from donor mice fed a chow diet or donor mice fed a HFD to recipient mice that were germ-free until colonized for this experiment. Recipient mice were all fed chow diet. After 7 days of daily exposure, feces were then transferred from donor to recipient mouse cages once per week. On Day 4 (**b**) and Day 45 (**c**) of microbiota transfer from donor to recipient mice the colonized germ-free, recipient mice were tested for glucose tolerance (N = 5, 5), where glucose tolerance test (GTT), area under the curve (AUC), and % body fat are shown. Statistical significance was measured with a Student t-test (**p < 0.01). Values are mean ± SEM. **d** PCoA of Bray-Curtis dissimilarity for all groups (left panel) and germ-free recipient groups only (right panel). **e** Boxplots showing the relative abundances of taxa that were significantly different between recipient mice that were exposed to feces from chow-fed or 60% HFD-fed donor mice. The Wilcoxon rank sum test was used to compare the two groups and calculate the significance (p < 0.05). Boxplots show median ± first and third quartiles. **f** Upset plot comparing taxa present in each group, where the y-axis shows the number of taxa common between the groups identified along the x-axis. Bar graph beside the x-axis shows the total number of taxa detected in each group

between donor (D) and recipient (R) mice (Fig. 8f). There were approximately 60 unique taxa within these mice and we found that 39 taxa were shared between all D and R mice, which is almost the exact same result as in the previous microbial transfer experiment (Fig. 7f). There were 13 taxa found in only a single group and 6 unique taxa that were shared between Chow_D and Chow_R groups. Six taxa were found in all groups of mice except the mice that ate an obesogenic diet (i.e. HFD_D group) and 4 taxa were found in both donor groups, but in only one of the recipient groups. These results are similar to those in Fig. 7, showing that the fecal microbiota transfer was successful in transferring most of the microbial communities from donor to recipient mice. The statistically different taxa observed in Fig. 7e and Fig. 8e were not identical. This discrepancy could be due to the time donors spent on HFD or the fact that the experiment conducted in Fig. 7 had a greater number of recipient mice, which likely strengthened the statistical test. Despite this difference, in both experiments it was a small number of taxa that significantly changed, which supports the conclusion that a small number of bacteria likely drive (or biomark) the glucose intolerance phenotype observed in the HFD recipient mice.

## Discussion

The microbiota has emerged as a factor in obesity, but less is known about how the microbiota could connect the progression of obesity to prediabetes and glycemic control. Three ill-defined concepts were: (1) the timing of changes in the constituents of the microbiota relative to the onset of obesity and glucose intolerance, (2) whether diet-induced changes in intestinal microbes could alter glycemia independently of altered adiposity, and (3) whether long or short term exposure to intestinal microbes from obesogenic diets contribute to dysglycemia.

It was known that HFD-feeding induces obesity and alters the gut microbiota[4,9–12]. However, it was still not clear if diet-induced changes in the microbiota precede glucose intolerance or vice versa. We found that both a 45 and 60% HFD rapidly altered the constituents of the microbiota, which preceded overt changes in glucose tolerance in mice. It was also known that dietary fat content related directly to the magnitude of changes in microbial taxonomy[12]. We did not isolate fat content in our dietary studies, but our results show that changes in the composition of the intestinal microbiota can be detected before changes in glycemia at both the onset and removal of obesogenic diets. Our data are consistent with a model where the presence of an obesogenic diet is the major factor influencing changes in fecal microbiota composition, rather than insulin resistance or dysglycemia altering the microbiota.

Previous work showed that 3–4 days of HFD feeding promotes glucose intolerance; however, both adipose tissue inflammation and lipid overload (independent of inflammation) have been proposed as mechanisms for glucose intolerance in response to this short-term feeding of obesogenic diets[13,16]. We have added

the timing of changes in the microbiota to this concept, but it remains to be determined if microbes participate in tissue inflammation or ectopic lipid disposition to an extent that is relevant to altered glucose control during short-term feeding of obesogenic diets.

Through fecal microbiota transfer experiments, we showed that the microbes associated with high fat feeding can alter glucose tolerance independent of adiposity. Our results are more consistent with microbiota contributing to glucose intolerance through metabolic inflammation during prolonged obesity, where the role of microbes in compartmentalized immune responses in the gut versus insulin responsive tissues should be carefully considered[17]. In particular, an attractive hypothesis is the requirement for certain obesogenic diets to alter components of the microbiota, which can act on the adaptive immune system to alter glycemia. Germ-free mice require approximately 40 days to generate an effective adaptive immune response upon exposure to commensal microbes[18,19]. This microbiota signal may be linked to metabolic inflammation in worsening glucose tolerance during long term obesity, where candidate pathobionts and metabolites have already been found[20,21]. The bi-phasic kinetics of changes in adiposity and glucose metabolism during colonization of germ-free mice with commensal bacteria have been documented. We sought to determine the effects of diet-induced obesity. It was already known that an early phase of colonization (during the first 3 days of exposure to microbes) associated with development of an inflammatory response, independent of changes in adiposity or glycemia. Impaired glucose control was only observed during the delayed phase of colonization that occurred between 14 and 28 days of exposure to microbes[22]. Further, it was shown that antibiotic treatment did alter the early phase of impaired glucose control[22]. Our results are consistent with these finding and we add to this model by showing that exposure time is a key factor where microbes or microbe-derived factors associated with diet-induced obesity potentiate late phase dysglycemia.

Given that long term exposure of mice to the microbes associated with an obesogenic diet was a key factor that contributed to glucose intolerance, our results question the utility of using fecal microbial taxonomy as a biomarker for glucose intolerance during diet-induced obesity since changes in taxonomy did not necessarily track with the manifestation of glucose intolerance. Our results warrant investigation of microbial metabolites that could gain access to host circulation or tissues to worsen glucose tolerance during long term obesity, likely acting in concert with host immune responses[15,23]. Further, investigation of diet-induced changes in microbial function and microbial-derived metabolites that can alter insulin resistance[24] may represent biomarkers compared to taxonomy. Nevertheless, our results provide insight into the mechanism of microbiota-driven changes in glucose tolerance.

It is not yet clear how to reconcile our results in mice with recent results showing that 7 days of Vancomycin or Amoxcillin

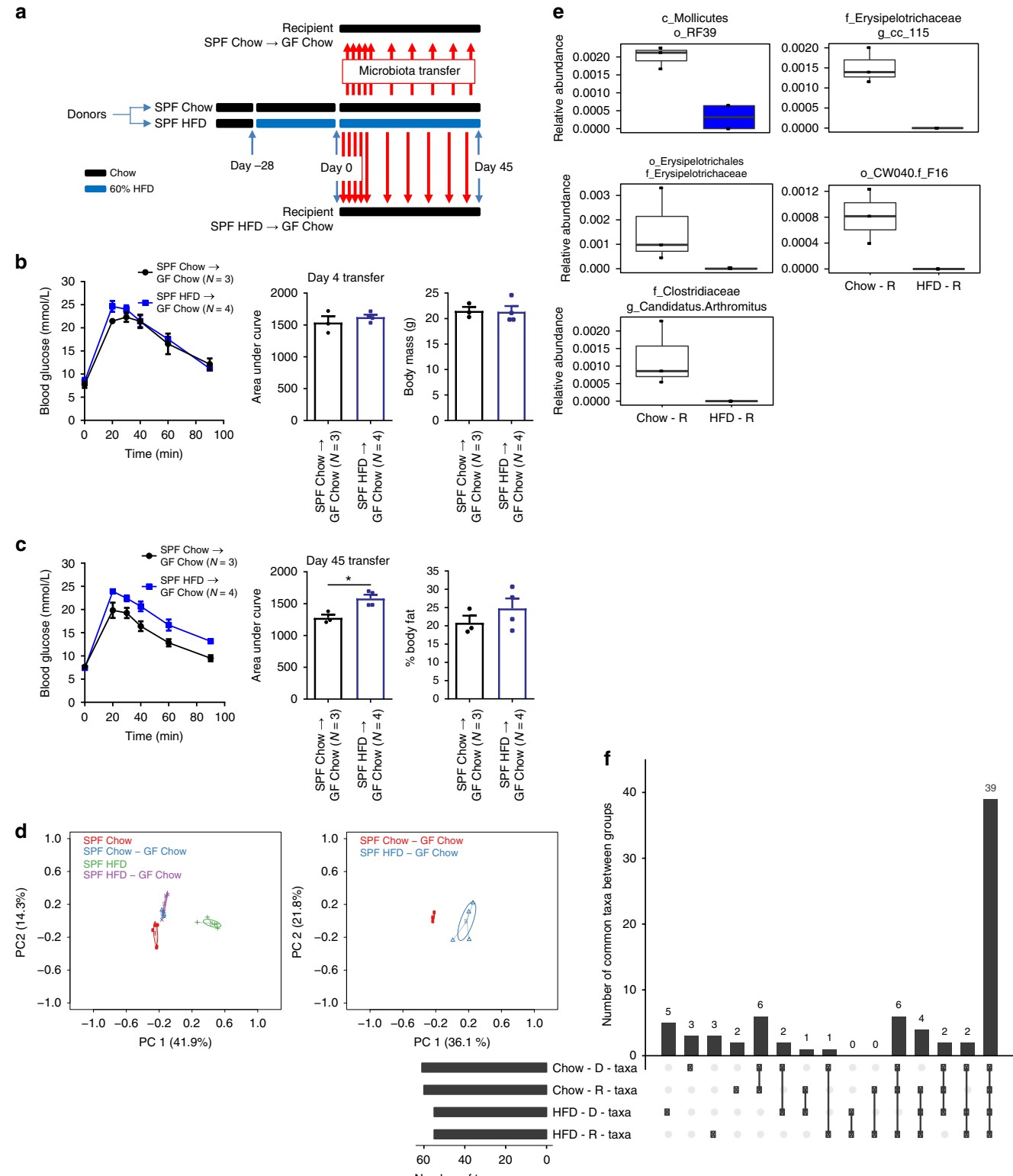

treatment had no impact on insulin sensitivity or substrate metabolism in obese humans[25]. The specific antibiotic used could be a key variable since (so far) we have found that only a combination of Ampicillin and Neomycin improves glucose tolerance in mice, an effect not seen with either antibiotic alone or with Vancomycin in HFD-fed mice. Other groups have also shown that specific combinations of antibiotics improve glucose tolerance in obese mice[15,26], but it is not yet clear if there is a

difference between mice and humans. Most importantly, given our results showing the requirement for prolonged exposure to microbes to alter glucose control in mice, the duration of exposure to obesogenic diets and altered microbial profiles/factors should be carefully considered in glycemic control of humans.

The constituents of the microbiota are modifiable, which may provide therapeutic targets in obesity and prediabetes. Our results show that diet-induced changes in the microbiota can

**Fig. 8** Long-term, but not short exposure to HFD-induced dysbiosis is sufficient for transmissible glucose intolerance, independent of changes in adiposity. **a** Schematic of experimental design. Specific pathogen free (SPF) donor mice were placed on chow or 60% HFD on Day -28. On Day 0, and each subsequent day, feces were transferred from donor mice fed chow diet or donor mice fed a HFD to recipient mice that were germ-free until colonized for this experiment. After 7 days of daily exposure, feces were then transferred from donor to recipient mouse cages once per week. On Day 4 (**b**) and Day 45 (**c**) of microbiota transfer colonized germ-free, recipient mice were tested for glucose tolerance ($N = 3, 4$), where glucose tolerance test (GTT), area under the curve (AUC), and % body fat are shown. Statistical significance was measured with a Student $t$-test (**$p < 0.01$). Values are mean ± SEM. **d** PCoA of Bray-Curtis dissimilarity for all groups (left panel) and germ-free recipient groups only (right panel). **e** Boxplots showing taxa that were significantly different between recipient mice that were exposed to feces from chow-fed or 60% HFD-fed donor mice. The Wilcoxon rank sum test ($p < 0.05$) was used to compare the two groups and calculate the significance ($p < 0.05$). Boxplots show median ± first and third quartiles. **f** Upset plot comparing taxa present in each group, where the $y$-axis shows the number of taxa common between the groups identified along the $x$-axis. Bar graph beside $x$-axis shows the total number of taxa detected in each group

influence glucose tolerance independently of obesity. Therefore, microbiota-based interventions such as prebiotics or probiotics may be able to lower glucose intolerance and insulin resistance separately from obesity. Our results showed that the duration of exposure to microbiota or microbiota-derived factors from obesogenic diets is a key factor dictating poor host glucose control. Over a month of exposure of the host to microbiota from obesogenic diets was required to worsen glycemic control, despite a rapid change in the taxonomy of the microbiota upon feeding these diets. Hence, exposure time is an important consideration in microbiota-targeted strategies aimed at attenuating dysglycemia versus adiposity.

## Methods

**Animal experiments**. All procedures were approved by McMaster University Animal Ethics Review Board. Specific pathogen free (SPF) *C57BL/6J* mice were born at McMaster University. Littermate mice were randomly placed on diets where: 45% HFD = fiber content of ~6%, 45% calories are derived from fat and the energy density is 4.7 kcal/g (Research Diets, D12451) or 60% HFD = fiber content of ~6%, 60% calories are derived from fat and the energy density is 5.21 kcal/g (Research Diets, D12492) or a chow diet containing 17% calories from fat and ~13% fiber content (Teklad 22/5 diet, catalog #8640). Blood glucose measurements and glucose tolerance tests were done after 6 h of fasting[14,27]. Mice were given glucose dose by intraperitoneal injection and blood glucose measures were taken by tail blood sampling at indicated times. Body fat composition was measured using whole body MRI (Bruker Minispec LF90-II). Antibiotics (1.0 mg/mL ampicillin and 0.5 mg/mL neomycin) were provided in the drinking water and changed every 2 days. Germ-free *C57BL/6N* mice were obtained from the Farncombe Gnotobiotic Unit of McMaster University and at 10–12 weeks of age. Upon export from the Gnotobiotic Unit, germ-free mice were immediately and continually colonized by housing mice in soiled litter from SPF *C57BL/6J* donor mice. Mice were individually housed using ventilated racks, and handled only in the level II biosafety hood to prevent bacterial contamination[14].

**Bacterial profiling**. Fecal samples were collected directly into sterile tubes and DNA was purified (Zymo Research Corporation: D4300)[14]. Following the mechanical disruption protocol outlined in the kit, we conduct 2 enzymatic lysis steps. First, 100 μL of lysis solution 1 (50 mg/mL lysozyme and 20% RNase—Sigma R6148) was added to each fecal sample and incubated at 37 degrees C for 1 h. Second, lysis solution 2 (25 μL of 25% SDS, 25 μL of 5 M NaCl, 50 μL of 10 mg/mL Proteinase K) was added to each fecal sample and incubated at 60 °C for 30 min. Illumina compatible PCR amplification of the variable 3 (V3) region of the 16 s rRNA gene was completed on each sample. The Illumina MiSeq platform was used to sequence DNA products of this PCR amplification. A minimum of 5696 reads per sample was acquired. A custom pipeline was used to process the FASTQ files[14,28]. Operational taxonomic units (OTUs) were grouped using Abundant OTU+ based on 97% similarity. The 2013 version of the Greengenes reference database was used to assigned taxonomy to OTUs Ribosomal Database Project (RDP) classifier in Quantitative Insights Into Microbial Ecology (QIIME). OTU assignments were converted to relative abundance before beta diversity calculations to account for depth of coverage and to normalize across samples. QIIME and R scripts were used to calculate beta diversity using the Bray-Curtis dissimilarity and principal coordinate analysis, to generate plots of taxonomy data, and to perform statistical tests[29–31]. Microbial taxonomy was expressed as relative abundance per sample. In heat maps, relative abundance was expressed as $\log_{10}$ fold change from the control group, as described in each figure. All relative abundance values of 0 were assigned $1 \times 10^{-7}$ in heat maps, the lowest detectable decimal value in the relative abundance, in order to allow the logarithmic transformation of the fold change. Statistical analyses were performed on relative abundance values. Then the $\log_{10}$ fold change from the control group for the most significantly changed taxa were plotted in a heatmap. In fecal microbiota transfer experiments, comparison of

taxa that changed between recipient groups was restricted to taxa that appeared in at least 60% of one of the recipient groups of mice.

**Statistical analysis**. For host metabolic measurements an unpaired, two-tailed Student's $t$-test was used to compare two groups and ANOVA and Tukey's post hoc analysis was used to compare more than two groups. Statistical significance was accepted at $p < 0.05$. Analysis and data visualization of microbial populations was conducted in R[30,31]. Partitioning of the variance in the microbiome was done with a Permutational multivariate analysis of variance (PERMANOVA) on Bray-Curtis dissimilarities calculated from relative OTU abundances, using the vegan package in R[32]. The Kruskal-Wallis test was used for the non-parametric analysis of variance between different groups (e.g., diets, timepoints) with the significance threshold set to $p < 0.05$. Subsequently, the Wilcoxon rank sum test was used for pairwise comparisons. Adjustment for the false discovery rate (FDR) was calculated with the Benjamini-Hochberg method[33] and statistical significance was accepted at $p < 0.05$.

**Code availability**. The custom R scripts used for data analysis are available from the corresponding author on reasonable request.

## Data availability

The datasets generated during the current study are available from the corresponding author on reasonable request. Figures that have associated raw data are: 2, 3, 4, 5, 6, 7, 8, and Supplementary figures 1, 2 and 3.

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

## Acknowledgements

K.P.F. was supported by an NSERC fellowship. B.D.M. was supported by Ontario Graduate Scholarships (OGS). Supported by a discovery grant to JDS from Natural Sciences and Engineering Research Council (NSERC) and a Foundation grant to JDS from the Canadian Institutes of Health Research (CIHR). JDS held CDA Scholar (SC-5–12–3891-JS) and CIHR New Investigator awards (MSH-136665) and holds a Tier 2 Canada Research Chair in Metabolic Inflammation.

## Author contributions

K.P.F. researched the data, contributed to the discussion, and edited the manuscript. S.Z. analyzed data and contributed to discussion. E.D., B.M.D., R.C. researched the data. J.C.S. analyzed data and contributed to discussion. J.D.S. researched the data, derived the hypothesis, wrote the manuscript and is the guarantor of this work.

## Additional information

**Competing interests:** The authors declare no competing interests.

