## [Peer Review File · Nature Communications]

Reviewers' comments:

Reviewer #1 (Remarks to the Author):

In the manuscript "Long-term dysbiosis promotes insulin resistance during obesity despite rapid diet-induced changes in the gut microbiome of mice" Foley et al investigates changes in gut microbiota composition during high fat diet and the causative role of the gut microbiota in relation to glucose homeostasis. By performing a series of in vivo experiment they show that changes in gut microbiota occur prior to changes in glucose resistance. They also show that impaired glucose intolerance can be transmitted by gut microbiota transfer from donors fed high fat diet for a prolonged period of time. In contrast, microbiota transfer from mice fed a high fat diet for a short period of time did not.

The studies are well designed, the outcomes are in most cases clear and the manuscript is well written. However, I have some questions that the authors will need to address prior to publication.

1. The authors should take into account the overall composition of the diets. E.g. the fiber content may be a significant contributor to the shaping of the gut microbiota.
2. The authors should account for diet consumption and put that in relation to weight gain and metabolic profile.
3. Since microbiota from mice fed HFD for 3 days and mice fed HFD for 14 weeks differ in the way they affect host metabolism it is of outmost importance that the microbiota composition and function at these time points are thoroughly compared. This comparison may help to identify the mechanisms underlying the influence on host phenotype.
4. Please show the BW at day 14 in figure 3 to make it possible to see shift in weight after food switch.
5. Please show the bacterial status of the mice after antibiotics treatment.
6. The microbiota in the recipient mice does not match the microbiota of the donor mice as illustrated in Figure 5e-f and Figure 6e-f. This needs to be thoroughly discussed.
7. The experiment in Figure 6 does not distinguish between a.) if the donors have to be on HFD for 28+45 days to have a microbiota that induces impaired glucose tolerance or b.) if the recipient mice need to be colonized for 45 days before the phenotype can be observed. To really show that it is the composition of the gut microbiota of mice fed HFD for 28+45 that induces these changes a study where donor mice are fed HFD for 28+45 days before the first colonization and then is subjected to a GTT after 4 days of colonization should be performed.
8. The discussion is mainly a repetition of the results from the present study. Please put some more emphasis on putting your results in relation to previous studies.

Reviewer #2 (Remarks to the Author):

The basic aim of the study is to clarify the order of events in term of physiological and microbiological responses to a high fat diet. The main claims revolve around a role for diet-induced change in microbiota in the onset of insulin resistance. I found the data interesting, but many aspects of the study (most notably the conclusions) were not clearly presented and it is difficult to see how the findings have actually progressed the field. The concluding sentence of the abstract does not really clarify the questions presented at the start of the abstract and if anything further confuses the issue.

I suspect there is a good story that would emerge with significant revision and further analysis.

A key issue is that some key concepts for presentation of the hypothesis are poorly framed. The

concept of dysbiosis is central to the study but ill-defined and inconsistently used. For example on line 65 "Gut dysbiosis during obesity is sufficient to increase adiposity...". Most would describe dysbiosis as a state of the whole system not a measure microbial community composition. If the system state is obesity, and this is measured as adiposity then this is doubly tautological. It would be clearer to say, transplant experiments show differences in a microbial community are sufficient to induce differences in adiposity when diet and genotype are controlled. Since dysbiosis is defined as an association between undesirable system state and distinct microbial community it is never going to be truly appropriate to refer to something happening within 1 to 2 days as dysbiosis (although 'early stages of establishment a dysbiotic state' would work).

Although the study aims to resolve short and long term effects of diet-induced obesity on insulin resistance there is no meaningful coverage of the literature in this regard. Of particular importance would be to consider the effects of dietary fat profile, inflammation and gradual increase in inflammophilic bacteria as defining features of long-term change (e.g Devkota et al. 2012 Nature; Lam et al 2015 Obesity).

The study design appears reasonable to address the questions but many aspects are only superficially described and some are overlooked.

There is no measurement of any aspect of previously described long-term changes. For example intestinal function (inflammation or permeability), systemic functions (inflammation, dyslipidemia) or fine resolution examination of microbial data to specifically test for previously observed marker organisms (*Bifidobacteria/Desulfovibionales* or *Enterobacteriaceae*).

The diets should be described in greater detail. The chow diet is not described anywhere - how can you consider it a control? Readers should be able to see basic details of the two HFD's without having to refer to a company web site. The comparison of diet do not appear to be well-controlled since the 45% fat diet contains starch and the 60% diet does not (which will lead to differences in the support of cecal microbiota). The differences in nutritional profile will also drive differences in palatability and feedback with the animals appetite regulation. You did not report intake, but I would expect both food and water intake to differ significantly between the diets. Are the differences in body weight and body fat reported after just one day artifacts of such intake changes?

On line 205 you say "We next reconstituted germ free mice" you may want to rephrase that! (Made me think of adding water to a powdered mix to get a mouse). These experiments are not clearly described in term of the protocol for microbe exposure, methods refer to continual housing with soiled litter on line 327, and oral gavage every 4 days on line 328 and then on line 209 its 'mice that received daily fees'. It is essential that this is clear since gavage with the feces of an animal with a bloom of inflammophilic microbes will give bioactive effects that are distinct from the colonisation effects.

The microbe analysis is rather superficial and not always presented in sufficient detail to support the findings. The basic data set and quality filtering were not described that I could see (How many reads per sample? How did you deal with differences in sample size or sample coverage?).

The use of stacked bar graphs to present average relative abundance is not adequate to support the community differences. You have relatively small numbers of animals and so showing all individuals in figures in supplementary data would be far more helpful.

The Phylum/Class level analyses are sufficient to show that community difference exists, but they are not really informative beyond that

The PICRUST analysis is so superficial as to be essentially meaningless.

You do show some interesting effects on Glucose tolerance between the treatments. These are shown at single time point (i.e. 2 days after the return to chow diet in Fig 3. Do you have a time course of GTT over longer periods? A longitudinal analysis of how both GTT and microbial

composition change with time would greatly add to the interest of the paper.

Reviewer #3 (Remarks to the Author):

This manuscript examines changes in the gut microbiome on mice in response to a HFD and discerns their role in HFD-induced obesity and dysglycemia. The central novel finding is that the microbiome plays a more important role in the long-term rather than short term dysglycemia induced by HFD. Moreover, the results indicate that, at least in this reviewer's understanding of the data, that it is long term exposure to the dysbiotic microbiota that promotes dysglycemia rather than that the dysbiosis need to drive dysglycemia requires long term exposure to HFD. Overall, this would be a solid conceptual advance in this field of crucial public health importance. Yet, the manuscript could certainly benefit from a clearer more concise writing style and some means of data presentation are not optimal. Specific comments follow:

1) The use of chow, 45%, and 60% fat diets to assess adiposity, dysglycemia, and microbiota composition is a very appealing model in that it provides a basis to assess the strong changes induced upon switch from chow to the highly obesogenic dysglycemia inducing diet and assess role of fat content per se but yet, just when it got interesting, namely that both induced similar degrees of dysbiosis but 60% induced more glycemia, the 45% dropped out of the manuscript. It would really be nice to see if the differences between 45% and 60% fat were ameliorated by antibiotics in the short and long term.

2) The way the microbiome is displayed is not particularly satisfying. It would really be preferable to see PCoA plots for each group over various days. At present, I believe the conclusion that dysbiosis precedes dysglycemia primarily based on data from Gary Wu and colleagues that HFD-induced dysglycemia is very rapid. It seems the sequencing data in the authors possession could make a better case using more broad based methods of presenting the data.

3) The text re the antibiotics is too strong. It seems clear that antibiotics have a much stronger effect on long-term dysglycemia than on short term but it seems highly likely there is still an impact on the latter regardless of whether it is or is not statistically significant in this experiment. Please soften tone to make less absolutist.

4) To this reviewer...The data in figure 5 suggests that either i) an aspect of dysbiosis not detected in sequencing is important for dysglycemia, or ii) long-term exposure the quickly occurring dysbiosis is needed to promote IR. The clever approach used in Figure 6 demonstrates the latter is correct. However, the text does not actually make these points in a coherent manner so I'm not sure if this is in fact the authors view of their data. If it is, they need to find a way to state it clearly, including in the abstract as this is one of the major advances of this work.

5) Given the cited work that long-term dysglycemia requires inflammation, this would seem a logical discussion point here.

Reviewers' comments:

Comment to all reviewers:

All reviewers commented that the writing clarity should be improved in order to deliver the most important scientific message. As such, we have completely rewritten the manuscript and not used "track changes".

Reviewer #1 (Remarks to the Author):

In the manuscript "Long-term dysbiosis promotes insulin resistance during obesity despite rapid diet-induced changes in the gut microbiome of mice" Foley et al investigates changes in gut microbiota composition during high fat diet and the causative role of the gut microbiota in relation to glucose homeostasis. By performing a series of in vivo experiment they show that changes in gut microbiota occur prior to changes in glucose resistance. They also show that impaired glucose intolerance can be transmitted by gut microbiota transfer from donors fed high fat diet for a prolonged period of time. In contrast, microbiota transfer from mice fed a high fat diet for a short period of time did not. The studies are well designed, the outcomes are in most cases clear and the manuscript is well written. However, I have some questions that the authors will need to address prior to publication.

1. The authors should take into account the overall composition of the diets. E.g. the fiber content may be a significant contributor to the shaping of the gut microbiota.

It was not the goal of our study to determine if fat % or any specific macronutrient or micronutrient was a participating factor in host physiology or microbiota changes. Accordingly, we have changed the text to now reflect that we used obesogenic diets. Although commonly identified based on their % fat content, these diets have many differences from the chow diet in terms of both caloric density and fiber content. We have done our best to highlight that the diets used are a model to induce obesity - an obesogenic diet that causes altered glucose tolerance and increase adiposity. This is associated with changes in the composition of microbiota, but we did not ascribe any feature to increased fat content. We have kept the routine nomenclature of "HFD" in certain places of the manuscript since this term is widely used and it is very onerous to re-explain all the differences of an obesogenic diet each time that is mentioned in the text. Of course, recent papers have addressed fiber composition and it most definitely will impact the microbiota composition (Zhao et al., Science 359, 1151–1156, 2018) and metabolic outcomes. All text that alluded to the diet phenotypes being driven by % fat has been re-worded accordingly.

2. The authors should account for diet consumption and put that in relation to weight gain and metabolic profile.

We have conducted new mouse experiments to address this point. We tracked food consumption for 7 days prior to switching diets. This included measuring food consumption each day of the first week of high fat feeding, and average consumption in the second week of high fat feeding. This is now Figure 1G. Mice fed the obesogenic diets consumed more food on the first day of high fat feeding, but this quickly corrected by Day 2. Despite similar food consumption of the two obesogenic diets on Day 1, only the more caloric dense diet, 60% HFD significantly increased adiposity after 1 day of feeding. Furthermore, gains in body mass and adiposity continue to increase after food consumption is reduced (when measured by per gram of food) in mice eating the obesogenic diets. These data suggest that increased food consumption alone does not cause the observed host phenotype.

3. Since microbiota from mice fed HFD for 3 days and mice fed HFD for 14 weeks differ in the way they affect host metabolism it is of outmost importance that the microbiota composition and function at these time points are thoroughly compared. This comparison may help to identify the mechanisms underlying the influence on host phenotype.

This is an excellent point raised by several reviewers. We conducted additional mouse-based experiments to compare the microbiota of mice fed chow, 45% HFD, or 60% HFD after 3 days or 14 weeks of feeding obesogenic diets. This is now Figure 3. We report that many of the changes in the composition of the microbiota that occur after 3 days of high fat feeding are similar after 14 weeks. The majority of the responses occur in unison regardless of the type of obesogenic diet (45% HFD versus 60% HFD) and regardless of the timing of the diets (3 days versus 14 weeks). However, there is a subset of taxa that differ between the two time points on obesogenic diets. Future studies may benefit from examining if these taxa can drive differential microbial effects on glucose metabolism during early versus long term obesity.

4. Please show the BW at day 14 in figure 3 to make it possible to see shift in weight after food switch.

Body mass on day 14 of 60% HFD is now reported in the text of the Results. There was no statistically significant change in body mass (33.4 versus 31.2 grams).

5. Please show the bacterial status of the mice after antibiotics treatment.

To address this comment, we conducted more animal based experiments. We re-performed the experiment with chow fed and 45% HFD fed mice. These data are now added into Figures 5 and 6 to show the effects of antibiotics treatment on the microbiota over the entire time course of these experiments. As expected, antibiotics alters the composition of the microbial taxonomy in the feces. We did not collect feces from mice fed chow or 60% HFD in this experiment –and we did not redo this experiment for comparison. We explain that any information from comparing antibiotic studies that are discordant for glucose metabolism will only generate associations between candidate taxa and glucose metabolism. Rather –this data set the stage for direct testing in germ free mice. We thought it redundant to go back and redo associative studies given the results in Fig 7 and 8.

6. The microbiota in the recipient mice does not match the microbiota of the donor mice as illustrated in Figure 5e-f and Figure 6e-f. This needs to be thoroughly discussed.

This is a valid point. We have expanded the analysis of all microbiota characterization, including the direct comparison in donor and recipient mice in both Figures 7 and 8. A more in depth analysis of the donor and recipient microbiota shows the differences between the two recipient groups and the “quality” of microbiota transfer between groups. The majority of microbial taxa were transferred between donor and recipient mice; however, the recipient microbiota may not have established with the exact relative abundances as observed in the donors. The stacked bar graphs that were presented only showed the relative abundance of the 12 most abundant taxa. Use of PCoA and Upset plots better depict the success of the microbiota transfer experiments. We did not expect extensive transfer of all the microbiota from donors to quantitatively similar levels in recipients given our “transmission” method using exposure to feces. When we see a phenotype using this method, we interpret it to be an advantage that the number of candidate taxa or bacterial derived molecules is reduced. The important point is that the host phenotype was transferred to recipient mice and that there was a difference in

glucose when mice (were exposed long enough) to the microbiota from HFD- donor mice versus chow fed mice. Our data comparing the constituents of the microbiota within recipient mice suggest that a relatively small number of taxa biomark the difference.

7. The experiment in Figure 6 does not distinguish between a.) if the donors have to be on HFD for 28+45 days to have a microbiota that induces impaired glucose tolerance or b.) if the recipient mice need to be colonized for 45 days before the phenotype can be observed. To really show that it is the composition of the gut microbiota of mice fed HFD for 28+45 that induces these changes a study where donor mice are fed HFD for 28+45 days before the first colonization and then is subjected to a GTT after 4 days of colonization should be performed.

We apologize for the confusion. We performed the required variation of this experiment between Figures 5 and 6 (now Figures 7 and 8). In Figure 5 (now Figure 7) we showed that 45 days of colonization was required to observe a defect in glucose tolerance when donor mice were placed on the HFD only 1 day before microbiota transfer. This raised the question as to whether the donor mice needed to be on HFD for 45 days or the recipient mice colonized for 45 days. In Figure 6 (now Figure 8), we placed the donor mice on HFD for 4 weeks prior to colonization. This experiment demonstrated that it still took 45 days of colonization to observe a defect in glucose tolerance. The experiment in Figure 6 was used to determine if it was the 45 days of colonization that was required or a longer exposure of the donors to diet. We found that even with donors being exposed to diet for longer, it took 45 days to observe the phenotype. Our data suggests that it is long term exposure to the microbes that is required to impact glucose metabolism. Please refer to comments from other reviewers (i.e. Reviewer #3) on these experiments.

8. The discussion is mainly a repetition of the results from the present study. Please put some more emphasis on putting your results in relation to previous studies.

We have re-written the discussion (and majority of the paper) to emphasize the significance of our findings relative to the current literature.

Reviewer #2 (Remarks to the Author):

The basic aim of the study is to clarify the order of events in term of physiological and microbiological responses to a high fat diet. The main claims revolve around a role for diet-induced change in microbiota in the onset of insulin resistance. I found the data interesting, but many aspects of the study (most notably the conclusions) were not clearly presented and it is difficult to see how the findings have actually progressed the field. The concluding sentence of the abstract does not really clarify the questions presented at the start of the abstract and if anything further confuses the issue.

I suspect there is a good story that would emerge with significant revision and further analysis.

A key issue is that some key concepts for presentation of the hypothesis are poorly framed. The concept of dysbiosis is central to the study but ill-defined and inconsistently used. For example on line 65 "Gut dysbiosis during obesity is sufficient to increase adiposity...". Most would describe dysbiosis as a state of the whole system not a measure microbial community composition. If the system state is obesity, and this is measured as adiposity then this is doubly tautological. It would be clearer to say, transplant experiments show differences in a microbial community are sufficient to induce differences in adiposity when diet and genotype are controlled. Since dysbiosis is defined as an association between undesirable system state and distinct microbial community it is never going to be truly appropriate to refer to something happening within 1 to 2 days as dysbiosis (although 'early stages of establishment a dysbiotic state' would work).

We have operationally defined what we mean by dysbiosis and re-phrased statements as appropriate. This is a fair point, but it is very difficult and onerous for the reader to continually re-state the state of the host and the state of the microbiota and whether each is adapted or not and actually pinpoint the time/characteristics when this has occurred. We have operationally defined dysbiosis and re-phrased statements as appropriate and removed dysbiosis as often as we could in the manuscript.

Although the study aims to resolve short and long term effects of diet-induced obesity on insulin resistance there is no meaningful coverage of the literature in this regard. Of particular importance would be to consider the effects of dietary fat profile, inflammation and gradual increase in inflammophilic bacteria as defining features of long-term change (e.g Devkota et al. 2012 Nature; Lam et al 2015 Obesity).

We have added to the discussion to address how our data may fit with current models of obesity induced glucose intolerance. We have included these and other recent and relevant citations in the discussion, which is now expanded to include more discussion of inflammation. We have clarified our use of obesogenic diets as a model to induce obesity and removed language that suggested dietary fat was an independent factor of changes in the microbiota or glucose tolerance. Hence, we did not specifically address dietary fat profiles as this was not the intent of the manuscript. We have improved our analysis of the microbiota such that readers may now identify all alterations to taxonomy with the diets over the time course of the study. Refer to next comment for specific examples of marker organisms that were identified in our analysis.

The study design appears reasonable to address the questions but many aspects are only superficially described and some are overlooked.

There is no measurement of any aspect of previously described long-term changes. For example

intestinal function (inflammation or permeability), systemic functions (inflammation, dyslipidemia) or fine resolution examination of microbial data to specifically test for previously observed marker organisms (*Bilophila/Desulfovibrionales* or *Enterobacteriaceae*).

We appreciate that we cannot make conclusions regarding inflammation, as we did not pursue this avenue of research. It exceeds the scope of this paper to harvest tissues at each of the time points examined and measure tissue inflammatory markers and serum lipids. This type of work would also produce correlative data rather than the ability to ascribe function of certain immune components in connecting microbe factors to glucose metabolism. We have explicitly detailed what we plan to do next in the discussion to address this very large concept. We will test the role of adaptive immunity in linking obesity-related microbes to glucose intolerance. This involves re-deriving several knockout mice under germ free conditions.

The presence of inflammation early during high fat feeding is established in the literature. In the discussion we have now addressed how our data may fit with that of inflammation to drive glucose intolerance during chronic obesity. We have measured intestinal permeability using FITC-dextran and found that 1 day of high fat feeding (60% HFD) increased intestinal permeability (unpublished data). However, this is outside the scope/focus of the current manuscript.

Importantly, we have built and implemented a new analysis pipeline for reporting microbiota results. Our new analysis of the microbiota identifies all taxa that significantly changed. In this analysis it can now be seen that taxa in the family *Enterobacteriaceae* are elevated on Day 3 of high fat feeding (Figure 2D) or Week 14 of high fat feeding (Figure 3C), and that one *Enterobacteriaceae* is reduced after 14 days of high fat feeding (Figure 4E). One *Bilophila* of the *Proteobacteria* phylum/*Desulfovibrionales* order was elevated in both HFDs on Day 3 and Week 14 of high fat feeding (Figure 3C). This *Bilophila* was also shown to be elevated by 60% HFD on Day 14, which remained elevated after 2 days of HFD removal (Figure 4E). We have avoided lengthy speculation on specific taxa in this manuscript.

The diets should be described in greater detail. The chow diet is not described anywhere - how can you consider it a control? Readers should be able to see basic details of the two HFD's without having to refer to a company web site. The comparison of diet do not appear to be well-controlled since the 45% fat diet contains starch and the 60% diet does not (which will lead to differences in the support of cecal microbiota). The differences in nutritional profile will also drive differences in palatability and feedback with the animals appetite regulation. You did not report intake, but I would expect both food and water intake to differ significantly between the diets. Are the differences in body weight and body fat reported after just one day artifacts of such intake changes?

We have described the diets in greater detail in the Results and Methods. The language used in the text now reflects that these are obesogenic diets. Although commonly identified based on their % fat content, these diets are different from the chow diet in both caloric content, fibre content and source of macro and micronutrients. All text that alluded to the diet phenotypes being driven by dietary fat percentage has been re-worded accordingly. The control diet was chosen as a commonly used chow diet that does not cause obesity. Despite differences in diet composition, the degree of adiposity and weight gain parallels caloric content of the diet (and calories obtained from dietary fat). We do not conclude what in the diet accounts for all changes in microbiota. We have added Figure 1G which tracks food consumption over the first 2 weeks of switching diets. The two HFDs show similar food intake, which does differ from that of chow. Our data suggests that gains in weight and body fat are not artifact – despite identical food consumption (measured by amount of food), mice fed 60% HFD increased adiposity faster than mice fed 45% HFD. Mice on both diets continued to gain weight and fat mass despite eating less food than the chow fed mice.

On line 205 you say "We next reconstituted germ free mice" you may want to rephrase that! (Made me think of adding water to a powdered mix to get a mouse). These experiments are not clearly described in term of the protocol for microbe exposure, methods refer to continual housing with soiled litter on line 327, and oral gavage every 4 days on line 328 and then on line 209 its 'mice that received daily fees'. It is essential that this is clear since gavage with the feces of an animal with a bloom of inflammophilic microbes will give bioactive effects that are distinct from the colonisation effects.

We have re-phrased "reconstituted" with "colonized". Experimental protocols have been clarified in the text. Detailed descriptions of colonization strategies are provided in the figure legends to accompany the schematics provided for experimental design. This is a good point since it is a point of the paper to determine the host responses to discordant timing of different microbe blooms.

The microbe analysis is rather superficial and not always presented in sufficient detail to support the findings. The basic data set and quality filtering were not described that I could see (How many reads per sample? How did you deal with differences in sample size or sample coverage?). The use of stacked bar graphs to present average relative abundance is not adequate to support the community differences. You have relatively small numbers of animals and so showing all individuals in figures in supplementary data would be far more helpful. The Phylum/Class level analyses are sufficient to show that community difference exists, but they are not really informative beyond that. The PiCRUST analysis is so superficial as to be essentially meaningless.

We have performed new analysis of the sequencing data for all figures to provide more in depth analysis. Visualizations beyond stacked bar graphs are presented to better highlight changes in taxonomy. PiCRUST analysis was removed completely to allow room for better coverage of taxonomy analysis. Additional sequencing information has been added to methods.

You do show some interesting effects on Glucose tolerance between the treatments. These are shown at single time point (i.e. 2 days after the return to chow diet in Fig 3. Do you have a time course of GTT over longer periods? A longitudinal analysis of how both GTT and microbial composition change with time would greatly add to the interest of the paper.

We have added a new figure – Figure 3 – that compares the microbial communities of day 3 versus week 14 fed mice. Comparing mice during the first week, second week and up to 14 weeks of dietary intervention is a long time course of longitudinal experiments. We did not pursue further experiments into diet removal – it was beyond the scope of this paper to map the time course for return to a complete chow-like microbiota and glucose tolerance. These experiments all set the stage for direct testing in germ free mice

Reviewer #3 (Remarks to the Author):

This manuscript examines changes in the gut microbiome on mice in response to a HFD and discerns their role in HFD-induced obesity and dysglycemia. The central novel finding is that the microbiome plays a more important role in the long-term rather than short term dysglycemia induced by HFD. Moreover, the results indicate that, at least in this reviewer's understanding of the data, that it is long term exposure to the dysbiotic microbiota that promotes dysglycemia rather than that the dysbiosis need to drive dysglycemia requires long term exposure to HFD. Overall, this would be a solid conceptual advance in this field of crucial public health importance. Yet, the manuscript could certainly benefit from a clearer more concise writing style and some means of data presentation are not optimal. Specific comments follow:

1) The use of chow, 45%, and 60% fat diets to assess adiposity, dysglycemia, and microbiota composition is a very appealing model in that it provides a basis to assess the strong changes induced upon switch from chow to the highly obesogenic dysglycemia inducing diet and assess role of fat content per se but yet, just when it got interesting, namely that both induced similar degrees of dysbiosis but 60% induced more glycemia, the 45% dropped out of the manuscript. It would really be nice to see if the differences between 45% and 60% fat were ameliorated by antibiotics in the short and long term.

We have conducted additional mouse-based experiments and now added chow versus 45% HFD to both short-term and long-term antibiotic models. These are now in Figure 5 and Figure 6.

2) The way the microbiome is displayed is not particularly satisfying. It would really be preferable to see PCoA plots for each group over various days. At present, I believe the conclusion that dysbiosis precedes dysglycemia primarily based on data from Gary Wu and colleagues that HFD-induced dysglycemia is very rapid. It seems the sequencing data in the authors possession could make a better case using more broad based methods of presenting the data.

This is a fair point. We have developed and implemented an improved analysis platform for the presentation of microbiota taxonomy in all figures. PCoA plots are provided in each figure for all groups, and when appropriate plots are provided for a specific sub-set of groups. Additional plots are provided to better depict the changes observed in the microbiota with high fat feeding, the changes that occur over time, and the quality of germ-free colonization. We tried to strike a balance to display results that are meaningful for "metabolism" focused researchers and microbiologists. We are happy to modify further –as suggested. All OTU data will be deposited publically.

3) The text re the antibiotics is too strong. It seems clear that antibiotics have a much stronger effect on long-term dysglycemia than on short term but it seems highly likely there is still an impact on the latter regardless of whether it is or is not statistically significant in this experiment. Please soften tone to make less absolutist.

This proved to be an excellent comment. We conducted additional experiments in chow fed and 45% HFD-fed mice. Results section for these experiments was re-written and conclusions changed to be more appropriate for the data described. The reviewer was correct that antibiotics have a stronger effect on long term dysglycemia, but some antibiotic-induced changes in glycemia can be seen during short term dysbiosis .

4) To this reviewer...The data in figure 5 suggests that either i) an aspect of dysbiosis not detected in sequencing is important for dysglycemia, or ii) long-term exposure the quickly occurring dysbiosis is

needed to promote IR. The clever approach used in Figure 6 demonstrates the latter is correct. However, the text does not actually make these points in a coherent manner so I'm not sure if this is in fact the authors view of their data. If it is, they need to find a way to state it clearly, including in the abstract as this is one of the major advances of this work.

Thank you for understanding the core concept in this paper. This indeed is our interpretation of the data as well. We have altered the language (and used the terms suggested by the reviewer) in the results and discussion to better highlight this finding and to place its significance in the context of the current literature.

5) Given the cited work that long-term dysglycemia requires inflammation, this would seem a logical discussion point here.

We have added discussion points to address the role of inflammation in short-term versus long-term obesity. We have placed our findings in the context of the literature and concluded that our findings support a model in which altered microbiota likely triggers metabolic inflammation during chronic obesity. The concepts of inflammation occurs mainly in the 4th and 5th paragraphs of the discussion. We propose that microbiota-induced adaptive immune responses are worthy of directly testing as a driver of insulin resistance.

Reviewers' comments:

Reviewer #1 (Remarks to the Author):

After the additional experiments performed and the changes made to the text I now think the ms is suited for publication.

Reviewer #2 (Remarks to the Author):

The manuscript has several different experiments that aim to distinguish the temporal role of microbiota-derived effects on the onset of diet-induced insulin resistance. These include short-term vs long term HFDs; the effect of switching from HD back to chow; the effect of antibiotic treatment and the effect of microbiota exposure. There is a lot of data here but much of it is either confirmatory (effects of diet on microbiota and IR) or phenomenological descriptions. I think the observations of the effect of antibiotic treatment on insulin resistance and of microbiological exposure are interesting. However, I found it difficult to see what new mechanistic insights arise from those observations and think the conclusions are overstated at best (and generally not clearly supported). In my view there is a lot of potential here, but the observations seem preliminary at present and interpretation of their significance would require further work.

Specific comments.

Regarding changes in body fat does the MRI infer body fat % at a tissue-specific level (esp liver) or is it the whole animal? If the former this would be more useful to report. If the latter, then was the MRI data normalised for what was probably a cecum full of high fat chyme?

The microbiome analyses are based on a large dataset but not always presented in the most informative way and often over-intrepreted. Most microbiota-related conclusions rely on rather simplistic interpretations of a PCA and a stacked bar chart. They support the basic story but I believe much more effective use of space could be made here. Figure 2 is essentially confirming numerous previously published studies and would be suitable as supplementary info.

Figure 3 would be of far more interest if the evident differences between the D3 and W14 communities were explored more meaningfully. Given you discuss several references (e.g. 13, 23, 24) that present a model for development of inflammation involving ongoing microbe changes this would be necessary to make any meaningful comparison of findings. It is potentially significant that your PCA show an effect of age in the chow-fed animals that is of greater magnitude, but different 'direction' in the ordination. Panel C is unclear to me. This would make more sense if all were normalized to day 0 chow (or day 3 chow if you don't have true baseline).

Figure 5 is hard to follow. I presume that the 19 animals in the HFD+Ab group represent two separate experiments since that is the only way the graphs in panel B make any sense. In the first panel here there are 19 data points for HFD-AB and they show significant difference in body mass to the 9 HFD animals. The next panel shows HFD+Ab compared to Chow+Ab but only shows 10 animals and these have a mean change in body fat of 6%. The final panel shows only 9 animals and they have a mean body fat change of 8%, which you conclude is not significantly different. It looks very much like there would be a difference if the other cohort were used.

The GF mouse 'exposure' experiments are conceptually fuzzy. The first day of exposure to feces is effectively colonization – they are no longer GF and should be referred to differently (perhaps 'conventionalised' or as 'gnotobiont'). In the context of metabolic disease the relevant exposure is in the intestinal tract and you have not done this in any controlled fashion. Did you consider that the mice may have consumed the feces and that the calorific content of HFD-fed feces may well have been significantly higher than chow-fed feces. Comparisons of fold change in rare taxa is not very meaningful and in my experience are seldom very reproducible. The SPF group in figs 7 and 8

are effectively an independent replicate – how similar are they? These are intriguing observations, but there is a very big gap between putting feces in a cage and what type of physiologically-relevant exposure to microbially-derived signals the animal tissues will see.

The methods should indicate how many independent cages of animals were in all experiments.

L34 You can't justify calling an 'immediate' diet-induced change in microbiota dysbiosis. Or at least if that is your definition of dysbiosis it is not very useful to understanding disease because it happens so frequently.

L95-97 This is unclear to me and I don't see how it would even be a useful concept since in the real world diet and microbiota are inextricably linked - it is impossible to separate the two. This needs to be defined far more clearly to be useful.

L115 I suspect these body mass changes are more likely to be explained by food/water intake (Fig 1G). I am a little surprised that food intake on the HFD diets was less than chow on all succeeding days. Is that typical in your animal house with those diets?

L197 This may not have been statistically significant at a cohort level but a 2 g in only 2 days is lot! Was it consistent across all animals if you track individuals? Also it implies they ate less on the chow (which I would expect) but is not consistent with what you said was the typical pattern on line 115.

Reviewer #3 (Remarks to the Author):

Manuscript is improved and makes nice contribution to field.

We were delighted that both Reviewers 1 and 3 only commented that:

"After the additional experiments performed and the changes made to the text I now think the ms is suited for publication."

"Manuscript is improved and makes nice contribution to field."

Comments from

Reviewer #2 (Remarks to the Author):

The manuscript has several different experiments that aim to distinguish the temporal role of microbiota-derived effects on the onset of diet-induced insulin resistance. These include short-term vs long term HFDs; the effect of switching from HD back to chow; the effect of antibiotic treatment and the effect of microbiota exposure. There is a lot of data here but much of it is either confirmatory (effects of diet on microbiota and IR) or phenomenological descriptions. I think the observations of the effect of antibiotic treatment on insulin resistance and of microbiological exposure are interesting. However, I found it difficult to see what new mechanistic insights arise from those observations and think the conclusions are overstated at best (and generally not clearly supported). In my view there is a lot of potential here, but the observations seem preliminary at present and interpretation of their significance would require further work.

Thank you for the careful review of this manuscript. We appreciate that it has been well documented that diet alters the microbiota and that this represents a factor that can impact insulin resistance. It is still not clear how this happens. In particular, the independent contributions of microbes to glucose metabolism are poorly characterized. This was a key, overall goal of our manuscript.

The reviewer is absolutely correct that previous work has already clearly established the timing of changes in intestinal bacteria after cycling obesogenic diets (i.e. Carmody ... Turnbaugh, *Cell Host Microbe*, 2015). We confirm these results and provide further evidence that it takes 3-4 days for diet to influence the majority of the taxonomic changes in the microbiota. Our manuscript then adds an advance to the existing literature beyond

confirmatory data because it fills a knowledge gap regarding the link to host metabolism. Namely, it was known and not been shown how the timing of changes in the microbiota during these first days of feeding an obesogenic diet compared to the timing with the onset of worse blood glucose control. We are the first to empirically show the relationship between changes in the microbiota and the onset of glucose intolerance. We can appreciate that the reviewer cannot see/determine a major advance in the initial timing and “antibiotic” studies in our manuscript. These experiments (and the comments above) merely set the stage for the most important experiments in our manuscript.

Most importantly our manuscript study goes on to characterize the duration of feeding an obesogenic diet that allows microbes impact glucose metabolism. This is the key advance of our manuscript and is presented in Figures 7+8. The major findings being that 1) microbes can alter glucose tolerance independent of ingested diet 2) long term exposure to these “obesogenic” microbes is necessary for the microbiota to impair glucose metabolism, 3) these effects can occur independently of worse obesity/adiposity. These are new mechanistic insights into how microbes alter metabolism. Critically, many previous studies showing partial, low magnitude or no transmission of glucose metabolism phenotypes to mice that were germ-free (but are then colonized) did not expose the recipient mice to the microbes for long enough to see changes in glucose tolerance – and in fact, often showed changes in fasting blood glucose or adiposity without showing an impairment in glucose tolerance. Inconsistencies between studies with regards to transmission of metabolic phenotypes to previously germ-free mice can be explained, at least in part, by differences in time of colonization/exposure. This information should be available to readers to plan the best experiments and provides insight into how microbes alter glucose control versus obesity.

Specific comments.

Regarding changes in body fat does the MRI infer body fat % at a tissue-specific level (esp liver) or is it the whole animal? If the former this would be more useful to report. If the latter, then was the MRI data normalised for what was probably a cecum full of high fat chyme?

The MRI that we used determines body fat in grams in the whole animal. The % body fat, lean mass and water content is calculated after weighing each mouse at the time of MRI, but this technique does not allow for segmentation of fat depots. MRI data is not normalized for the presence of chime in the cecum. It was not possible to fast these mice repeatedly in order to minimize what could be influence from dietary fat and cecal chime because MRI measures were taken every day and repeated fasting would significantly lower body weight and body fat.

The microbiome analyses are based on a large dataset but not always presented in the most informative way and often over-interpreted. Most microbiota-related conclusions rely on rather simplistic interpretations of a PCA and a stacked bar chart. They support the basic story but I believe much more effective use of space could be made here. Figure 2 is essentially confirming numerous previously published studies and would be suitable as supplementary info.

We agree that the influence of diet to rapidly alter the microbiota has been well documented. However, the purpose of Figure 2 is to highlight the specific differences in microbial taxa in this study on day 3 after switching diets – the day before diet induced impairment in glucose tolerance. It was not the goal to determine a specific taxa that discriminated obesity or any other metabolic factor. Rather we wanted to display the altered bacterial community in response to diet in the absence of major changes in glycemia. This was effectively done to Day 3 after changing the chow diet to a HFD. Essentially the use of PCoA and stacked bar graphs validate why we chose Day 3 to then conduct a more in depth analysis of the microbiota. We chose to include these simpler interpretations of the data to guide readers toward the logic of choosing specific time point and more in depth analysis of microbiota depicted in heat maps and Supplemental bar graphs. Figure 2 also sets up further comparisons into how early changes in the microbiota differ from those that occur after long term feeding of HFD. It may be very difficult for readers to follow the logic of the paper if we remove Fig 2.

In order to address the reviewers concern, we have added an additional comments that specifically states that our results in Fig 2 confirm previous work. We have included after commenting on Fig 2 in the results: "These results are consistent with previous reports documenting the rapid effect of diet on gut microbial taxa¹⁶."

Figure 3 would be of far more interest if the evident differences between the D3 and W14 communities were explored more meaningfully. Given you discuss several references (e.g. 13, 23, 24) that present a model for development of inflammation involving ongoing microbe changes this would be necessary to make any meaningful comparison of findings. It is potentially significant that your PCA show an effect of age in the chow-fed animals that is of greater magnitude, but different 'direction' in the ordination. Panel C is unclear to me. This would make more sense if all were normalized to day 0 chow (or day 3 chow if you don't have true baseline).

Our intent in discussing the role of inflammation in contributing to microbiota induced glucose intolerance was to place our results in the context of current literature. It was specifically requested by multiple reviewers in the previous review of our manuscript to discuss this point regarding inflammation and current literature. However, additional experiments to *define* the role of inflammation are well outside the scope of this paper. It is our opinion that it is not particularly useful to *describe* the associations with inflammation by adding measurements of a few or even comprehensive inflammatory markers at different time points during HFD feeding. Our data provides a useful starting point for future endeavors into whether or not these changes in microbiota from Day 3 to Week 14 contribute to inflammation and the progression of glucose intolerance. However, it is also plausible that the change in microbiota that occurs by Day 3 of high fat feeding is sufficient to cause inflammation and glucose intolerance if present for a sufficient amount of time. Our contribution here is to characterize the microbial community over time during exposure to HFD – which spring boards us into subsequent figures that assess whether these microbes can independently alter glucose metabolism. Future work will focus on defining the role of specific immune responses by re-deriving several "immune response deficient" mice under germ-free status in our gnotobiotic unit and exposing these mice to

obesogenic microbes. This will take several years, but appears more informative than measuring some markers of inflammation.

For Figure 3, Panel C, the data presentation was selected in order to highlight contribution of aging from our analysis. This was requested from reviewers. Initially the goal was to observe if changes in microbiota taxonomy on Day 3 (HFD relative to Chow) versus those observed on Week 14 (HFD relative to Chow) were of similar magnitude. This was achieved. However, if we express the fold change relative to Day 3 chow for all groups, as now requested by the reviewer, the data in manuscript would conflate the effects of diet and those of aging. Particularly now that we know that aging alters the microbiome in our samples, this will hinder the point we are attempting to address and actually be less transparent.

Figure 5 is hard to follow. I presume that the 19 animals in the HFD+Ab group represent two separate experiments since that is the only way the graphs in panel B make any sense. In the first panel here there are 19 data points for HFD-AB and they show significant difference in body mass to the 9 HFD animals. The next panel shows HFD+Ab compared to Chow+Ab but only shows 10 animals and these have a mean change in body fat of 6%. The final panel shows only 9 animals and they have a mean body fat change of 8%, which you conclude is not significantly different. It looks very much like there would be a difference if the other cohort were used.

The reviewer is correct. Two separate experiments were performed: 1) to compare chow vs HFD while using antibiotics and 2) to compare HFD vs HFD+Ab. We have now combined the experiments for panel B for showing change in % body fat and adjusted the text in the manuscript accordingly. It now reads:

“Despite small reduction in body mass and fat mass gains, antibiotics did not prevent increased adiposity during this short-term 60% HFD-feeding (Fig. 5B).”

The GF mouse ‘exposure’ experiments are conceptually fuzzy. The first day of exposure to feces is effectively colonization – they are no longer GF and should be referred to differently (perhaps ‘conventionalised’ or as ‘gnotobiont’).

We agree that upon export the germ-free mice are no longer germ free. It is critical to note in manuscripts that mice were born germ-free as this model has advantages and disadvantages. We always note when mice were born germ free so the reader understand that this model was used. The “previously germ free” mice are routinely referred to in our manuscript and these mice colonized immediately upon export from the gnotobiotic unit and selective pressure for colonization with microbes is maintained by continual exposure to donor feces. We state this method in the first sentence of the Results section describing the colonization experiments. Continual reference to “germ-free mice” is for simplicity to identify the groups of mice – they are never described as being tested under germ-free status. They are referred to as “colonized” or “recipient” or “previously” germ-free mice.

In the context of metabolic disease the relevant exposure is in the intestinal tract and you have not done this in any controlled fashion.

The method of colonizing germ-free mice with soiled cages from donor mice is a well-established method for altering the intestinal microbiota that we and others have published previously. We have tested many different methods of colonizing germ free mice, including those routinely used in the literature such as gavage of cecal contents among others. There are pros and cons to each method. Over many years we have found that cage exposure to feces can sufficiently promote continual selective pressure with minimal stress to the recipient mice, which promotes effective measurement of metabolic responses such as glucose control –that are particularly sensitive to stress. We have published this method before we think it is an important factor for other groups to consider.

For this manuscript specifically, we made no attempt to conclude about the species/strains of bacteria or location of action to alter glucose metabolism. The relevant exposure may be in the intestinal tract, a certain component of the intestinal tract or some other body site. It is not known. The most important factor is that we transmitted a measureable phenotype with our method. We used our method to show that transfer of microbes can alter glucose metabolism and our analysis of fecal microbes provides insight into candidate taxa for biomarking the metabolic phenotype. Specific details

regarding exact locus of action within the intestinal tract is beyond the scope of this study.

Did you consider that the mice may have consumed the feces and that the calorific content of HFD-fed feces may well have been significantly higher than chow-fed feces.

This is true. It is likely that there is some caloric content in the HFD feces that is remnant of the original diet. However, given that impaired glucose tolerance is transmissible in the absence of obesity or increased adiposity (Figure 8) and that the germ-free recipient mice exposed to donor feces from HFD donors did not cluster with their donor groups in the PCoA plot, it is very unlikely that remnants of the HFD in the feces given to the germ-free recipient mice caused the microbial changes or metabolic phenotype observed.

Comparisons of fold change in rare taxa is not very meaningful and in my experience are seldom very reproducible.

The comparison of fold change in rare taxa was required to show a summary of the microbiota analysis for several reasons. The magnitude of changes in relative abundance of taxa dictated displaying the data as fold change in a heatmap. This is common when datasets have both abundant and rare factors and when the goal is to look for signatures of change. This heatmap of fold change was the only meaningful method of conveying the result in a figure for readers. The manuscript would be far less transparent if we remove this data. Importantly, we have included the quantification of relative abundance in supplemental figures (S2, S3 and S4). All of the data is there for inspection and readers who want to see more data than just fold change.

We appreciate the reviewers experience with rare taxa and use fold change in rare taxa. The supplemental figures specifically address the fold change issue – since all data is presented if readers are interested. The reviewers experience with rare taxa is shared by us (and other labs) in terms of reproducibility. However, we must balance potential noise or false positive results with importance and transparency of data. It has become apparent that microbes or taxa observed to be of low relative abundance are not necessarily of low

importance and can influence on host response, including metabolism. We have conducted additional analyses of our data. First, we want to highlight some in built check and balances of our experimental design and existing data. To address the reviewers concern about reproducibility in the current study we want to be clear that we examined the effect of 3 days of obesogenic diet on changes in the taxa of microbiota in 2 independent experiments (Figures 2 and 3). These were separate groups of mice and experiments were done months apart. We reported similar changes in taxonomy in both experiments. This was done by design in order to address reproducibility.

We now have performed a multitude of additional analysis to rigorously test how various filters alter the rare taxa that are reported. First, and most importantly was a filter that was already present in the manuscript. We only present the data that is significantly different (by non-parametric testing) between several groups of mice. It is very unlikely that “sequencing noise” would be reliably detected for specific taxa in multiple groups of mice –and be consistently different across diets and times of dietary changes such that it produced a statistically significant result. Nevertheless, it is still possible that some rare taxa are spurious results. We and others contend that low abundance does not equate to low importance, hence we do not want to discard useful data. Here in the response to reviewers we applied data filters to see how our rare tax data is changed. These results are below. We already applied a filter to sequencing data from Figure 7+8, where samples had to be present in 75% of at least one treatment group. We have now applied this and even more stringent filters to Figure 2 –as an example for review.

In our assessment our data is largely unchanged after passing it through stringent filters to reduce false positives. We are emboldened/justified by the following figures, which show that the sequencing data remain quite robust to “culling” with increasing cut-off stringencies. **We have chosen to only present this data here in the response to reviewers. If absolutely required by the editor and reviewer - we can apply these filters to data in the paper – possibly in another supplemental figure. We think showing it here demonstrates that this is not needed or even useful to convey the message of the paper.**

Data:

Figure 2D modified from manuscript. We applied one filter to test rare taxa. We found that 25 Taxa remained present when data must be present in 75% samples in at least one sample group. The original figure had 36 taxa.

Figure 2D modified from manuscript. We applied two filters to test rare taxa. We found that 22 Taxa remained present when data must contain > 10 read counts in 75% samples in at least one sample group. The original figure had 36 taxa.

Figure 2D modified from the manuscript. Shown is the original heatmap without filters. It contained 36 significantly changed taxa. Taxa in red+purple remain in the list when one filter is applied (i.e. 25 taxa present in 75% samples in at least one treatment group). Taxa in purple fall out when 2 filters are applied (> 10 read counts in 75% samples in at least one sample group) leaving 22 taxa in the heatmap.

Figure 2D from the manuscript. Shown is the original heatmap without filters. It contained 36 significantly changed taxa. Taxa in red+purple+green remain in the list when one filter is applied (25 taxa present in 75% samples in at least one treatment group). Taxa in purple fall out when 2 filters are applied (> 10 read counts in 75% samples in at least one sample group). 21 taxa in green remain when the bottom 25% of reads are dropped from the analysis

The SPF group in figs 7 and 8 are effectively an independent replicate – how similar are they? These are intriguing observations, but there is a very big gap between putting feces in a cage and what type of physiologically-relevant exposure to microbially-derived signals the animal tissues will see.

The experiments in Fig 7 and 8 show a striking similarity. This is best depicted in the upset plots (i.e. Fig 7F and Fig 8F). It is stringing that 38 taxa are shared between donors and recipients in Figure 7 and 39 taxa are shared between donors and recipients in Figure 8. This is the key comparison rather than comparing spf groups which is not that informative since the spf (i.e. donor) mice were on the HFD for different duration (i.e. 45 days versus 45+28 days). This comparison is redundant to the initial figures of the paper that describe the timing of changes in taxa during an obesogenic diet.

Each method of colonizing germ-free mice with microbes have their advantages and disadvantages. Please refer to the aforementioned comments (above) regarding the method of exposing/colonizing germ free mice. We have a lot of experience with these gnotobiotic and microbial transfer methods inherited from the Farncombe Institute members and learned in our lab since we have tested many methods of microbial transmission for altering glucose metabolism. It is a common misconception that other methods such as gavaging cecal contents or other intestinal-focussed methods are superior or more rigorous. The continual pressure and level of stress must be considered. Most importantly our ability to show transmission of a metabolic phenotype independent of direct exposure of the host to the HFD suggests that the physiologically relevant bacteria or bacterial/host factors were transferred to the germ-free recipients. Any method of colonizing mice with microbes can be questioned regarding a physiologically-relevant exposure. In short, our method was highly successful regarding an exposure that induced an effect. Many researchers appear to be focusses on “extent” of transfer of the taxa from donors to recipient mice. However, it is an advantage is an experiment transfers a phenotype, but has a low penetrance or invasiveness of transferring microbial taxa.

The methods should indicate how many independent cages of animals were in all experiments.

Germ-free mice were exported from the gnotobiotic unit and individually housed. This is now clarified in the methods.

The methods for Animal experiments now reads
"Mice were individually housed using ventilated racks, and handled only in the level II biosafety hood to prevent bacterial contamination"

L34 You can't justify calling an 'immediate' diet-induced change in microbiota dysbiosis. Or at least if that is your definition of dysbiosis it is not very useful to understanding disease because it happens so frequently.

We have changed the word "dysbiosis" on L34. We have changed this to "Microbial taxa were altered". This should alleviate the reviewers concern about semantics and the use of the word "dysbiosis".

The reviewers point is valid, but not shared by all. We still contend that dysbiosis can be used to capture this "short-term" change. We believe that it is clearer for the reader to define this altered microbiota as dysbiosis as opposed to choosing an arbitrary length of time before calling the same microbial population dysbiosis or not. The same principal could be applied to any physiological effect and related term, including glucose intolerance, insulin resistance or obesity.

L95-97 This is unclear to me and I don't see how it would even be a useful concept since in the real world diet and microbiota are inextricably linked - it is impossible to separate the two. This needs to be defined far more clearly to be useful.

This is a fair point. It is true that the intestinal microbiota and the diet are "inextricably linked".

We can imagine how the following statement conflates diet and microbes without defining the important advance by our paper.

“Our results support a model where a sufficient exposure time of the host to the microbiota present during an obesogenic diet, rather than continued feeding of an obesogenic diet, is required to promote host dysglycemia.”

We have reworded the statement to define the useful concept.

It now reads

“Our results support a model where sufficient exposure time of the host to the microbiota-derived factors present during an obesogenic diet is a factor that permits microbes to contribute to poor glucose control. Our data support the concept that host exposure time is a key factor to consider in the development of dysglycemia and warrant caution in the assumption that that continual evolution of the microbiota during long-term feeding of an obesogenic diet is required for poor host glucose control. This time required for microbe factors to promote dysglycemia should be considered independent from obesity and despite rapid diet-induced changes in the microbiota.”

L115 I suspect these body mass changes are more likely to be explained by food/water intake (Fig 1G). I am a little surprised that food intake on the HFD diets was less than chow on all succeeding days. Is that typical in your animal house with those diets?

We have repeatedly observed decreased food intake in animals on the HFD within days of diet switch. This response is the typical response to this Research Diets sourced HFD in C57/Bl6J mice and is seen by many labs at McMaster (i.e. Dr. Gregory Steinberg) and other labs outside our institution. Given our observation that food intake was not higher in the HFD-fed mice, but these animals still gained more weight and adiposity over the first week of high fat feeding suggests that increased food intake is not the sole explanation for increased body mass.

L197 This may not have been statistically significant at a cohort level but a 2 g in only 2 days is lot! Was it consistent across all animals if you track individuals? Also it implies they ate less on the chow (which I would expect) but is not consistent with what you said was the typical pattern on line 115.

A decrease of 2 g was not statistically different. Even without eating less, there is less caloric content per gram in chow diet versus HFD. The loss of 2 g body mass in the 2 days of diet removal likely reflects this change in caloric content of the diet. It is plausible that the mice ate less, as their caloric consumption rate was adjusted to eating HFD. Upon removing a HFD, it generally takes a few days for the mice to adjust eating patterns to a new diet. We show in Figure 1G that the mice on HFD ate less than chow fed controls by Day 4. It may take a day or two for the HFD animals to reset their food consumption upon switching back to chow diet. This is not clear, but the (lack of) results of HFD removal back to a chow diet on glycemia are very clear.

The key points here are: This small loss in body mass in mice switched from a HFD to a chow diet was consistent across all animals (although not significant by statistical testing). Most importantly, despite losing approximately 2 g and switching to chow diet, the group of mice that was previously fed HFD remained glucose intolerant. A significant decrease in body mass would only help to improve glucose tolerance. Hence, a small weight loss effect – as described by the reviewer would only work to make our findings more difficult to observe. If the mice weighted less, the mice should be *less* glucose intolerant. The whole point was that these mice stayed glucose intolerant despite the dietary change. Even given all of this, the weight loss did not reach statistical significance.

We thank the reviewer and editor for a rigours review and taking the time to understand our manuscript.

REVIEWERS' COMMENTS:

Reviewer #4 (Remarks to the Author):

I think that this manuscript is ready for publication and it will be a great resource paper for the field. It is not necessary to include the additional figures into the supplementary section. The data presented in the manuscript are sufficient.

A minor comment- while the links between microbial related inflammation and insulin resistance are emerging, this manuscript does not contain any readouts on host immune responses. I understand why you brought up inflammation in the discussion but it is misleading to include "inflammation" as one of the keywords. Perhaps it's best to leave it out.

Reviewer #4 comments:

I think that this manuscript is ready for publication and it will be a great resource paper for the field. It is not necessary to include the additional figures into the supplementary section. The data presented in the manuscript are sufficient. A minor comment- while the links between microbial related inflammation and insulin resistance are emerging, this manuscript does not contain any readouts on host immune responses.

I understand why you brought up inflammation in the discussion but it is misleading to include "inflammation" as one of the keywords. Perhaps it's best to leave it out.

We thank the reviewer for their supporting comments. We have removed "inflammation" from the key words.